



# Effects of water table level and nitrogen deposition on methane and nitrous oxide emissions in an alpine peatland

**Wantong Zhang[1,2,4], Zhengyi Hu[2], Joachim Audet[4], Thomas A. Davidson[4], Enze Kang[1,3], Xiaoming Kang[1,3], Yong Li[1,3], Xiaodong Zhang[1,3], Jinzhi Wang[1, 3, *]**

[1]Institute of Wetland Research, Chinese Academy of Forestry, Beijing Key Laboratory of Wetland Services and Restoration, Beijing 100091, China

[2]Sino-Danish Centre for Education and Research, University of Chinese Academy of Sciences, Beijing 100049, China

[3]Sichuan Zoige Wetland Ecosystem Research Station, Tibetan Autonomous Prefecture of Aba 624500, China

[4]Department of Ecoscience and Arctic Research Centre (ARC), Aarhus University, Vejlsøvej 25, 8600 Silkeborg, Denmark

***Correspondence to**: Jinzhi Wang (wangjz04@126.com)

**Abstract**

Alpine peatlands are recognized as a major natural contributor to the budgets of atmospheric methane ($CH_4$) but as a weak nitrous oxide ($N_2O$) source. Anthropogenic activities and climate change have put these fragile nitrogen (N)-limited peatlands under pressure by altering water table (WT) levels and enhancing N deposition. The response of greenhouse gas (GHG) emissions from these peatlands to these twin changes is uncertain. To address this knowledge gap, we conducted a mesocosm experiment in 2018 and 2019 investigating individual and interactive effects of three WT levels ($WT_{-30}$, 30 cm below soil surface; $WT_0$, soil-water interface; $WT_{10}$, 10 cm above soil surface) and multiple levels of N deposition (0, 20, 40, 80 and 160 kg N ha$^{-1}$ yr$^{-1}$) on growing season $CH_4$ and $N_2O$ emissions in the Zoige alpine peatland, Qinghai-Tibetan Plateau. We found that the elevated WT levels increased $CH_4$ emission, while the N deposition had non-linear effects (stimulation at moderate levels and inhibition at higher). In contrast no clear pattern of the effect of WT levels on the cumulative $N_2O$ emission was evident, while N deposition led to a consistent and linear increase (emission factor: 2.3%-2.8% and 1% in IPCC), and this was dependent on the WT levels. Across the two years, the scenario with the greatest GWP (from $CH_4$ and $N_2O$) was an N deposition of ca. 20 kg N ha$^{-1}$ yr$^{-1}$ and high WT levels (at soil surface or above). Given the current N deposition in the Zoige alpine peatland (1.08-17.81 kg N·ha$^{-1}$), our results suggested that the $CH_4$ and $N_2O$ emissions from the alpine peatlands could greatly increase in response to the possible doubling N deposition in the future. We believe that our results provide insights into how interactions between climate change and human disturbance will alter GHG emissions from this globally important habitat.

**Keywords** Cumulative GHG emissions; multi-level N enrichment; critical threshold; non-linear effect; Qinghai-Tibetan Plateau

## 1. Introduction

Peatlands only cover ca. 3% of the land surface of the Earth but store one third of the global carbon pool (Yu et al. 2010).



In pristine peatlands, the shallow water table (WT) and waterlogged conditions allow accumulation of organic matter and
favour anaerobic production of methane ($CH_4$) and nitrous oxide ($N_2O$). Traditionally, the large carbon pool is nitrogen
deficient and is recognized as a moderate $CH_4$ source and a weak $N_2O$ source (Frolking et al. 2011, Han et al. 2019).
Nevertheless, these conditions could be markedly changed by anthropogenic disturbance and climate change, and growing
evidence shows that peatlands are experiencing drainage and increasing nitrogen deposition (Chen et al. 2013, Evans et al.
2021). Consequently, the magnitude of $CH_4$ and $N_2O$ emissions from peatlands is potentially changing from low to high. The
high-altitude or alpine peatlands are of particular interest because this carbon-rich ecosystem is vulnerable and highly sensitive
to the climate change and anthropogenic activities (Squeo et al. 2006).

Large-scale artificial drainage of peatlands was initiated hundreds of years ago and escalated in the 20[th] century (Evans

et al. 2021). As a result, about 10-20% of the global peatlands were primarily drained for the purposes of agriculture, peat
extraction and forestry (Frolking et al. 2011). The resulting lower WT altered the anaerobic conditions of the peat soil and led
to oxidative loss of peat (Laine et al. 2019, Wilson et al. 2016). Generally, the drainage decreased the $CH_4$ efflux and increased
$N_2O$ emissions (Cao et al. 2017). The increase in $N_2O$ emissions from drained peatlands is small, but may potentially reach a
high level at sufficient nutrient input, especially when the soil is fertilized (Laine et al. 2019). Ecological restoration has been
proposed as a measure to conserve the drained or degraded peatlands, particularly in order to meet the demand for mitigation
of GHG emissions outlined in the Paris Agreement (Evans et al. 2021). Numerous studies have reported a remarkably decreased
$CO_2$ effflux in rewetted or restored peatlands, but the rising WT levels have also augmented the emissions of $CH_4$ and $N_2O$
(Audet et al. 2013, Järveoja et al. 2016).

Atmospheric N deposition, primarily caused by anthopogenic activities (i.e. fossil fuels combustion, fertilizer application),

has increased consistently during the past decades (Gomez-Casanovas et al. 2016, IPCC 2013), and it is predicted to increase
two- or three-fold in terrestial ecosystems by the end of the century (Lamarque 2005). The increasing N deposition could
alleviate the N stress on peatlands, but the N effects on $CH_4$ and $N_2O$ emissions are controversial (Deng et al. 2019). Thus,
positive (Juutinen et al. 2018), negative (Gao et al. 2014) or neutral (Wang et al. 2017) effects of N deposition on $CH_4$ emissions
in peatlands have been observed. We speculate that the contrasting results probably are a result of the prevailing environmental
conditons and the N addition rate. Besides $CH_4$ emison, N deposition generally stimulates $N_2O$ emissions from peatlands due
to the increasing supply of N substrate (Wang et al. 2017). However, previous studies have also shown that a higher N input
leads to a transition of the grassland into a state of declining N saturation as well as a reduction of the sensitivity of the GHG
exchange to the continuously increasing N deposition (Gomez-Casanovas et al. 2016). To eliminate the possible gap resulting
from the N addition rate, multiple levels of N deposition are required to study the possible linear or non-linear effects of
deposition on GHG emissions.



Numerous studies have reported on the individual effects of WT and N deposition on GHG emissions in peatlands (Evans
et al. 2021, Saiz et al. 2021). To our knowledge, only a few studies (Gao et al. 2014, Wang et al. 2017) exist that focus on their
interactive effects on peatland GHG emissions. Gao et al. (2014) found that N addition in peatlands decreased $CH_4$ emissions
but increased $N_2O$ emissions without any siginificant interaction with WT levels. Wang et al. (2017) observed no interactive
effects of a lower WT and increasing N deposition on GHG emissions in an alpine wetland. The above-mentioned studies were,
however, limited to a single level of N addition and associated water addition. The response of GHG emissions in peatlands to
the gradients of N deposition and WT levels remains to be elucidated, in particular at the N saturation stage, even though it
may be a key factor in shaping GHG emissions. The large uncertainties regarding the interactive effects of N deposition and
WT levels on GHG emissions severely hamper obtaining a reliable estimation of the reponse of peatlands to climate change
and anthropogenic activities.
To address this knowledge gap we conducted a mesocosm investigation to study the influence of three WT levels (from
drained to inundated) and multi-level N deposition (from non-addition to 160 kg N ha$^{-1}$ yr$^{-1}$) on the soil $CH_4$ and $N_2O$ emissions
in the Zoige alpine peatland, located on the eastern edge of the Qinghai-Tibetan Plateau. Being the largest and highest swamp
wetland area in China, its sensitivity to the global climate change and human activities is high (Chen et al. 2013). Exposure to
a potential influence of drainage, restoration or increasing N deposition (Yang et al. 2017, Zhang et al. 2011) may increase the
risk of high GHG emissions from this area. In this study, we hypothesised that i) a slight increase in N deposition might
stimulate both $CH_4$ and $N_2O$ emissions, but a highlevel would inhibit GHG emissions. ii) The effects of N deposition on $CH_4$
and $N_2O$ emissions would be associated with WT levels due to the influence of WT-induced aerobic conditions on the efficiency
of utilising nutrients for $CH_4$ and $N_2O$ production.
**2. Methods and Materials**
**2.1 Study site**
This study was conducted in the Zoige alpine wetland, situated on the eastern edge of the Qinghai–Tibetan Plateau,
Southeast China, during the 2018 and 2019 growing seasons. This alpine wetland covers an area of 6180 km$^2$, which is 31.5%
of the whole Zoige plateau. The mean annual temperature is 1.4°C, with a maximum of 9.1 to 11.4°C in July and a minimum
of -8.2 to -10.6°C in January, while the average annual precipitation is approximately 650 mm (Chen et al. 2013, Yang et al.
2014). Over the past four decades, the mean annual air temperature has increased by 0.4°C per decade, while the total annual
precipitation has decreased by 22 mm per decade (Chen et al. 2013, Yang et al. 2014). Data on precipitation and air temperature
in this study were obtained from the closest meteorological station belonging to the Chinese National Meteorological
Information Center (www.nmic.gov.cn) and are shown in Figure S1. The depth of peat in the vertical profile is around 1.2 m,



soil pH is 6.8-7.2 and soil bulk density around 0.78 g m$^{-3}$ (Zhang et al. 2020). The plant growing season ranges from July to
September, and the dominant plants are *Carex mulienses*, *Lancea tibetica*, *Potentilla anserina* L. and *Trollius farreri Stapf.*
**2.2 Experimental design**
Our experiment was carried out at the Sichuan Zoige Wetland Ecosystem Research Station, Tibetan Autonomous
Prefecture of Aba (33°57′N, 102°52′E, 3500 m a.s.l.). A homogeneous swamp wetland was selected for collection of soil and
plants to be used in the mesocosm. Forty-five tanks (0.6 m length × 0.6 m width × 0.6 m height) were kept above ground and
filled with intact soil cores and vegetation. The bottom of the tanks was welded, and the outside of the tanks was wrapped with
polystyrene foam to avoid heat exchange with the surroundings.
The experimental treatments consisted of five levels of added N and three water table levels and applied in a factorial
design (5 N addition × 3 water table). The treatments were replicated three times, giving a total of 45 experimental plots. Based
on previous studies indicating water levels effects on GHG emission in the Zoige peatland (Cui et al. 2017, Yang et al. 2017),
three water table levels (WT$_{-30}$,- 30 cm below the soil surface; WT$_0$, 0 cm at the soil-water interface; and WT$_{10}$, 10 cm above
the soil surface) were selected. To maintain the water table at the selected three levels, we developed a water table control
system composed of three water table detectors, a manostat, a relay and micropumps. Three water table detectors were placed
in the PVC pipe (diameter 3 cm) of each tank at the exact water table level and at +2 cm and -2 cm water table. When the
water table was below the -2 cm detector, the pump switched on, supplying the tanks with local tap water until the water table
reached the +2 cm detector. To prevent the water table from becoming too high due to pump water or precipitation, four small
holes (diameter 1 cm, and two holes for two sides) were drilled at the precise position of the water table in each tank to allow
water overflow.
The current N deposition in the Zoige area is 1.08-17.81 kg N·ha$^{-1}$·yr$^{-1}$, NH$_4^+$ and NO$_3^-$being the main component, and N
deposition is expected to increase further in the future (Han et al. 2019). NH$_4$NO$_3$ was adopted as N source to simulate the
different stages of the response of alpine peatlands to multi-level N deposition, and five N addition levels were established for
each water table level, namely 0 (N$_0$), 20 (N$_{20}$), 40 (N$_{40}$), 80 (N$_{80}$), and 160 (N$_{160}$) kg N ha$^{-1}$ yr$^{-1}$. The annual added N doses
were further divided into four portions and applied at the beginning of every month from June to September in 2018 and 2019.
25% of the added N was dissolved into 1 L water and sprayed evenly upon the surface of each plot. while the control plot only
received 1 L water (Wang et al. 2017).
**2.3 GHG sampling and measurements**
We measured CH$_4$ and N$_2$O fluxes once a month from June to October in 2018 and two or three times a month from June
to September in 2019. In each tank, CH$_4$ and N$_2$O fluxes were measured using static opaque chambers and gas chromatography



(GC) (Zhang et al. 2017). The cubic chamber was made of stainless steel (0.5 m length × 0.5 m width × 0.5 m height; without
bottom). At the top surface of the chamber, there were two ports for headspace gas sampling and enclosed air temperature
measurements, respectively. A dry battery-powered fan was placed in the chamber to avoid stratification of the gases during
sampling. Meanwhile, 45 square collars (0.5 m length × 0.5 m width × 0.2 m height) were produced and buried into the middle
part of the soil core in each tank at about 0.4 m depth. Before placing the chambers on top of the collars to collect gas samples,
the collars were sealed with water to ensure minimum gas exchange between chamber and atmosphere.
Gas samples (20 mL) were collected from each chamber using a rubber tube connected to the valve of the chamber and
a syringe at 10 min intervals over a 20-minute period (0, 10 and 20 min). The samples were then injected into a pre-evacuated
10 ml vacuum vial (Aladdin, Shanghai, China). The samples were kept cold and dark until the $CH_4$ and $N_2O$ fluxes were
determined via GC (Agilent 7890A, Agilent Co., Santa Clara, CA, USA) within 72 hours. The GC was equipped with a flame
ionization detector (FID) to analyse the $CH_4$ concentration and an electron capture detector (ECD) to analyse the $N_2O$
concentration. The carrier gas was $N_2$, and the operation temperature for the FID was set at 250 °C and ECD at 300 °C.
The $CH_4$ and $N_2O$ fluxes were calculated by the slopes of linear regression between gas concentration and sampling time
(0, 10, 20 min after chamber closure). Each linear regression was assessed individually, and the estimates of the data quality
of the fluxes were uniformly dependent on R-squared values. However, the coefficients of determinations ($R^2$) of the linear
regression for $CH_4$ and $N_2O$ were sometimes low (<0.4), particularly when the fluxes were low. We did not want to create bias
against these low fluxes and therefore kept them if the $CO_2$ concentration (data not shown) showed a good linear trend with
time (Laine et al. 2019). Apart from these fluxes, values were generally accepted only if the R-squared values of the linear
regressions were equal to or greater than 0.8 and 0.7 for $CH_4$ and $N_2O$ (Lafuente et al. 2020, Laine et al. 2019), respectively.
The $CH_4$ flux had 5.29% discarded values, while the $N_2O$ flux had 3.70% discarded values.
**2.4 Analysis of soil properties**
To determine soil properties, soil samples were collected in late September, considered as the end of the growing seasons
in 2018 and 2019. Three sub-samples of soil were collected from each tank at the top 5 cm depth and then bulked into a
composite sample representing a reliable replicate. The collected soil samples were stored under cold and dark conditions and
then forwarded to the laboratory within three days. The soil samples were passed through a 2 mm sieve and air dried for the
determination of soil pH, soil organic carbon (SOC) and total nitrogen (TN). Soil pH was measured at a soil:water ratio of
1:2.5 with a pH electrode (PHS 29, China). SOC and TN were determined via dry combustion using an Elementar Vario Max
CN analyzer (Hanau, Germany). Soil water content (SWC) was determined by using a TDR300 moisture meter (Spectrum
Technologies Inc., Plainfield, Illinois, USA).





**2.5 Statistical analysis**

Generalized least square (GLS) ANOVA was used to assess the effect of WT and N (fixed factor) on the cumulative $CH_4$

and $N_2O$ emissions in 2018 and in 2019 (respectively), and the soil properties were determined via the R package *nlme* (Pekár
et al. 2016, Tiemeyer et al. 2016). We also used the GLS method to compare the effects of N deposition on $CH_4$ and $N_2O$
emissions at each WT level in each year, followed by a Tukey HSD test. The GLS model included an autoregressive structure,
accommodated for unequal time of sampling, and a variance function allowing for unequal variance in the fixed factors
(Wanyama et al. 2019).

A Generalized Additive Model (GAM) was used to fit the relationship between the cumulative $CH_4$ emissions and N

deposition dosages at different water table levels as well as the relationship between the global warming potential (GWP) of
growing season GHG emissions and N deposition dosages. Compared to the linear models, GAM directly and smoothly fitted
the non-linear relationship between the response variable and the multiple explanatory variables, despite the data distribution
(Chen et al. 2021).

The cumulative GHG emissions in the growing seasons in each year were calculated by the following equation:

Cumulative $CH_4$ (or $N_2O$) emission =

$$F_1 \times (t_1 - t_{start}) \times 24 + \sum_{i=1}^{n}(F_i + F_{i+1})/2 \times (t_{i+1} - t_i) \times 24 + F_n \times (t_{end} - t_n) \times 24$$
where F is the $CH_4$ (g C $m^{-2}$ $h^{-1}$) and $N_2O$ flux (g N $m^{-2}$ $h^{-1}$), n is the total number of measurements each year, $F_1$ and $F_n$ stand
for the GHG flux from the first and last sampling each year, and $t_1$ and $t_n$ represent the time of the first and last sampling each
year. i is the ith measurement, $(t_{i+1} - t_i)$ is the days between two adjacent measurements, $t_{start}$ and $t_{end}$ are 1 June and 30 September
each year. To reduce the heterogeneity of the first and last sampling occasion each year, the GHG emission from 1 June to the
first sampling and from the last sampling to 30 September was taken into consideration and compared to the results of previous
studies (Goldberg et al. 2010).

GWP was used to define the cumulative impacts of the growing season GHG budgets ($CH_4$ and $N_2O$) as a time-integrated

radiative force over a period of 100 years. The GWP of cumulative $CH_4$ and $N_2O$ emissions was calculated using the formula:

GWP (g $CO_2$-eq $m^{-2}$) = $28 \times F_{CH4-C} \times 16/12 + 265 \times F_{N2O-N} \times 44/28$

where $F_{CH4-C}$ and $F_{N2O-N}$ are the growing season cumulative emissions of $CH_4$ (g $CH_4$-C $m^{-2}$ $yr^{-1}$) and $N_2O$ (g $N_2O$-N $m^{-2}$ $yr^{-1}$)
based on the mass of C and N, respectively. The radiative forcings of $CH_4$ and $N_2O$ are 28 and 265 in terms of a $CO_2$-eq unit
at a 100-year time horizon.

Statistical analysis was carried out by applying the statistic R software (version 3.4.3) (R Development Core Team, 2011),

and graphs were drawn using OriginPro 9.8.0.200. Final *p* values were Bonferroni adjusted to mitigate the risk of type I error.
A significance level of $\alpha$ = 0.05 was used for all statistical tests.



**3. Results**
**3.1 Soil properties**
During the two years of the growing season mesocosm experiment, the soil water content (SWC) varied from 63.5% to
81.1% and was only significantly affected by the water table levels (Table 1 and 2). The higher WT levels significantly
increased the SWC in both years. The soil pH varied within the range 7.3 to 7.8 and was only significantly affected by N
deposition. Large variability of SOC was observed within the range 215.9 g kg$^{-1}$ to 296.1 g kg$^{-1}$, and both the WT and N
treatments showed significant effects on SOC, without any interactive effects. Compared with the control treatment without N
deposition, N deposition increased SOC by 1.4% to 31.1% (except $WT_0 N_{160}$ in 2019). Soil TN varied between 12.9 g kg$^{-1}$ and
19.1 g kg$^{-1}$ and was elevated by N deposition, whereas no significant response to the WT treatments was observed. N deposition
enhanced soil TN by 1.3% to 110.5% compared to the $N_0$ treatment at each WT level.

Table 1. Soil properties (mean ± SE) (n=3) in the different water table (WT) treatments and nitrogen deposition (N) levels in
2018 and 2019.

| | | 2018 | | | 2019 | | |
|---|---|---|---|---|---|---|---|
| | | $WT_{-30}$ | $WT_0$ | $WT_{10}$ | $WT_{-30}$ | $WT_0$ | $WT_{10}$ |
| SWC | $N_0$ | 65.8±2.5 | 67.4±2.6 | 69.8±1.5 | 62.8±1.7 | 70.4±1.0 | 81.1±1.6 |
| (%) | $N_{20}$ | 66.8±1.8 | 74.2±1.6 | 74.2±1.9 | 63.5±2.0 | 71.3±0.4 | 79.0±2.0 |
| | $N_{40}$ | 67.3±1.5 | 73.2±3.0 | 71.0±1.7 | 64.4±1.9 | 71.6±1.9 | 79.8±1.6 |
| | $N_{80}$ | 67.4±2.2 | 72.0±0.8 | 73.1±1.1 | 67.8±0.8 | 69.6±1.4 | 77.4±2.1 |
| | $N_{160}$ | 64.4±1.3 | 68.0±2.2 | 72.8±1.8 | 68.1±0.7 | 72.2±1.8 | 81.1±1 |
| | | | | | | | |
| pH | $N_0$ | 7.6±0 | 7.7±0.1 | 7.7±0.1 | 7.7±0 | 7.8±0.1 | 7.8±0.1 |
| | $N_{20}$ | 7.5±0 | 7.7±0 | 7.4±0.1 | 7.7±0.1 | 7.5±0.2 | 7.6±0.1 |
| | $N_{40}$ | 7.3±0 | 7.6±0.1 | 7.6±0.1 | 7.5±0.1 | 7.6±0.2 | 7.7±0.2 |
| | $N_{80}$ | 7.6±0 | 7.4±0.1 | 7.5±0.1 | 7.4±0.1 | 7.6±0.1 | 7.4±0 |
| | $N_{160}$ | 7.5±0 | 7.6±0 | 7.3±0.1 | 7.5±0.1 | 7.4±0.1 | 7.5±0 |
| | | | | | | | |
| SOC | $N_0$ | 231.3±5.4 | 237±24.3 | 246.6±16.0 | 215.9±3.6 | 227.3±14.4 | 218.3±14.0 |
| (g kg$^{-1}$) | $N_{20}$ | 296.1±5.7 | 285.8±8.4 | 279.2±23.4 | 228.7±9.6 | 249.9±12.0 | 273.3±11.8 |
| | $N_{40}$ | 292.3±14.1 | 281.2±18.7 | 262.8±20.9 | 241.8±6.7 | 281.0±17.8 | 253.3±17.5 |
| | $N_{80}$ | 265.4±17.7 | 294.3±7.7 | 291.4±9.3 | 240.9±12 | 230.6±10.8 | 286.2±9.1 |
| | $N_{160}$ | 275.6±7.0 | 276.8±10.1 | 266.8±32.4 | 254.6±18.2 | 226.8±13.7 | 251.5±19.1 |
| | | | | | | | |
| TN | $N_0$ | 17.6±0.8 | 16.1±1.1 | 18.7±0.8 | 12.9±0.9 | 14.4±0.3 | 14.7±2.1 |
| (g kg$^{-1}$) | $N_{20}$ | 18.7±0.7 | 19.1±0.6 | 19.3±0.8 | 21.9±2.2 | 21.2±3.2 | 23.3±5.0 |
| | $N_{40}$ | 18.4±1.1 | 18.8±0.9 | 19.2±0.4 | 19.1±1.5 | 21.4±4.0 | 15.0±2.7 |





| | | | | | | |
|---|---|---|---|---|---|---|
| $N_{80}$ | 18.3±0.8 | 19.4±0.2 | 19.7±0.2 | 18.6±1.1 | 16.2±1.0 | 31.0±2.6 |
| $N_{160}$ | 18.1±0.8 | 18.2±0.4 | 19.0±0.4 | 19.5±2.0 | 21.7±6.6 | 21.9±4.6 |

SWC, soil water content; SOC, soil organic carbon; TN, total nitrogen.

Table 2. Effects of water table (WT) and nitrogen (N) deposition levels and their interactions on soil properties using generalized least square (GLS) models.

| | SWC | | pH | | SOC | | TN | |
|---|---|---|---|---|---|---|---|---|
| | F | $P$ | F | $P$ | F | $P$ | F | $P$ |
| WT | 19.4 | **<0.001\*\*\*** | 0.34 | 0.7103 | 9.92 | **<0.001\*\*\*** | 2.08 | 0.1319 |
| N | 0.64 | 0.6352 | 6.78 | **<0.001\*\*\*** | 5.18 | **0.001\*\*** | 4.49 | **0.002\*\*** |
| WT×N | 0.25 | 0.9807 | 0.35 | 0.944 | 0.91 | 0.5147 | 0.74 | 0.6526 |

Bold font denotes significant values. The statistical significance levels used were: *: $0.01 < P < 0.05$; **: $0.001 < P < 0.01$; ***: $P < 0.001$. SWC: soil water content; SOC: soil organic carbon; TN: total nitrogen.

### 3.2 Methane

The Zoige alpine peatland acted as a net source of $CH_4$ in the $WT_0$ and $WT_{10}$ treatments throughout the two growing seasons, although the $CH_4$ flux was almost 0 in the $WT_{-30}$ treatment. Seasonal variability of the $CH_4$ flux was observed (Figure 1). The cumulative $CH_4$ emissions of the growing season ranged from -0.26 to 29.26 g $CH_4$-C $m^{-2}$ $yr^{-1}$ in 2018 and from -0.35 to 16.36 g $CH_4$-C $m^{-2}$ $yr^{-1}$ in 2019 (Figure 2). During the two years, the WT treatments and their interaction with the N treatments had significant effects on the cumulative $CH_4$ emissions, while N deposition only had significant effects in 2019 (Table 3). Along the WT level gradient from $WT_{-30}$ to $WT_{10}$, the cumulative $CH_4$ emissions increased markedly. The response of the cumulative $CH_4$ emissions to N deposition was non-linear under $WT_0$ and $WT_{10}$ conditions (Figure 3), the highest value occurring in the $N_{20}$ treatment. Compared to the $N_0$ treatment, the $N_{80}$ and $N_{160}$ treatments remarkably decreased the cumulative $CH_4$ emissions by 36.5% to 97.4%, while $N_{40}$ was in the same order of magnitude as in the $N_0$ treatment. The GAM results showed that the cumulative $CH_4$ emissions could be explained by N deposition for 55.9% under $WT_0$ conditions and for 45.4% under $WT_{10}$ conditions. The modelling results also indicated that the critical thresholds for the highest cumulative $CH_4$ emissions were 14.41 g C $m^{-2}$ $yr^{-1}$ with 20.9 kg $ha^{-1}$ $yr^{-1}$ N deposition under $WT_0$ conditions and 21.60 g C $m^{-2}$ $yr^{-1}$ with 16.2 kg $ha^{-1}$ $yr^{-1}$ N deposition under $WT_{10}$ conditions.

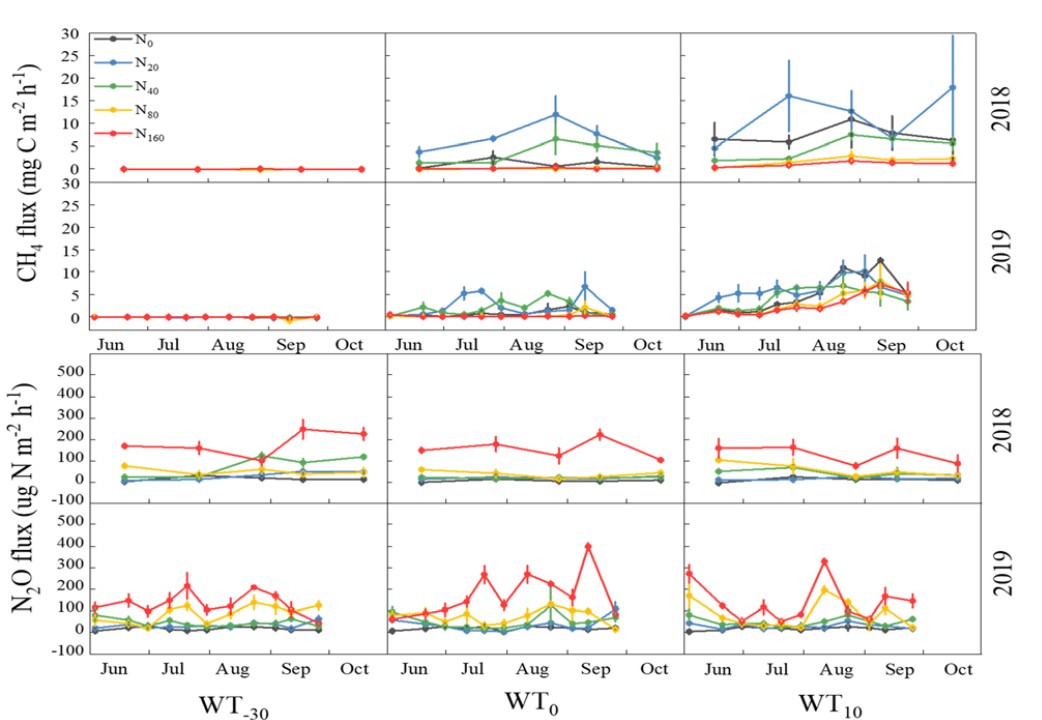

Figure 1. Temporal variation of the response of $CH_4$ and $N_2O$ fluxes to nitrogen deposition at three water table levels during the growing seasons in 2018 and 2019. Error bars represent the SE (n=3).

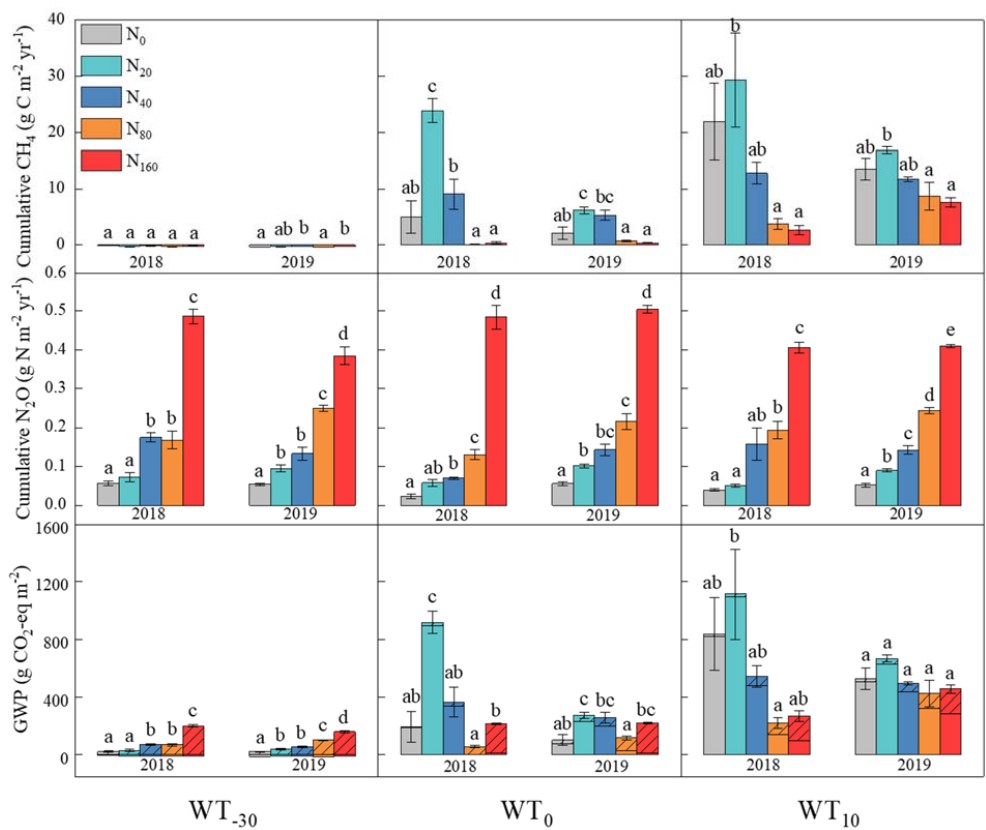

Figure 2. Effects of nitrogen deposition levels on cumulative $CH_4$, $N_2O$ emissions and GWP (from $CH_4$ and $N_2O$) at three
water table levels during the rowing seasons in 2018 and 2019. For the GWP figures, the shaded area indicates GWP generated
from cumulative $N_2O$, while the unshaded area from $CH_4$. Error bars represent the SE (n=3). Different letters above the bars
indicate statistically significant differences ($P < 0.05$).

Table 3. The individual and interactive effects of water table (WT) and nitrogen (N) deposition levels on GWP (global warming
potential), cumulative $CH_4$ and $N_2O$ emissions in 2018 and 2019 using generalized least square (GLS) models.

| | | $CH_4$ | | $N_2O$ | | GWP | |
|---|---|---|---|---|---|---|---|
| | | F | *P* | F | *P* | F | *P* |
| **2018** | | | | | | | |
| | WT | 24.88 | **<0.001***** | 36.68 | **<0.001***** | 177.63 | **<0.001***** |
| | N | 1.37 | 0.27 | 239.38 | **<0.001***** | 103.82 | **<0.001***** |
| | WT×N | 15.15 | **<0.001***** | 4.28 | **0.002**** | 13.59 | **<0.001***** |
| | | | | | | | |
| **2019** | | | | | | | |
| | WT | 615.89 | **<0.001***** | 351.26 | **<0.001***** | 702.04 | **<0.001***** |
| | N | 5.99 | **0.001**** | 989.75 | **<0.001***** | 66.67 | **<0.001***** |





| WT×N | 18.01 | **<0.001***** | 5.23 | **<0.001***** | 7.61 | **<0.001***** |
| --- | --- | --- | --- | --- | --- | --- |

Bold font denotes significant values. The statistical significance levels used were: *:0.01<$P$ < 0.05; **: 0.001< $P$ < 0.01; ***:
$P$<0.001.

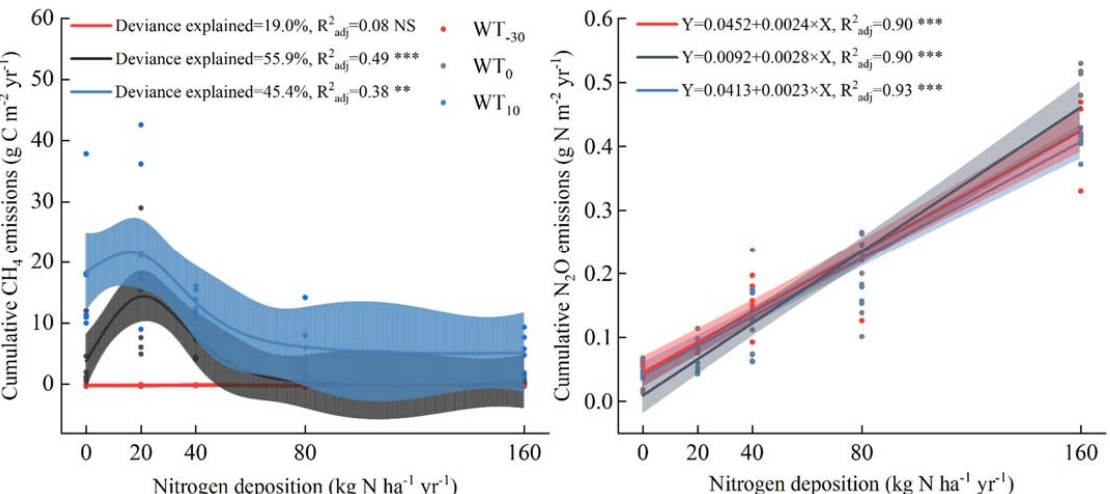

Figure 3. The relationship between cumulative greenhouse gas ($CH_4$ and $N_2O$) emissions and nitrogen deposition levels at
three water table levels. A linear model was used to estimate cumulative $N_2O$ emission at five nitrogen deposition levels,
while a generalized additive model (GAM) was used to assess the response of cumulative $CH_4$ emission to nitrogen
deposition levels. The statistical significance levels used were: * = $P$ < 0.05 and > 0.01; ** = $P$ < 0.01 and > 0.001; ***
$P$<0.001; NS = not significant ($P$ > 0.05). The shaded areas indicate 95% confidence intervals.

**3.3 Nitrous oxide**

The Zoige alpine peatland acted as a net $N_2O$ source during the growing seasons, the $N_2O$ fluxes showing clear seasonal

variability in 2018 and 2019. The $N_2O$ flux tended to be highest in early September 2018 and in mid-August 2019, while the
lowest flux was observed at the start or the end of the growing seasons (Figure 1). The cumulative $N_2O$ emissions ranged from
0.02 to 0.49 g $N_2O$-N m$^{-2}$ yr$^{-1}$ in 2018 and from 0.05 to 0.50 g $N_2O$-N m$^{-2}$ yr$^{-1}$ in 2019. The cumulative $N_2O$ emissions were
significantly affected by the WT levels, N deposition and their combination (Table 3). N deposition significantly increased the
cumulative $N_2O$ emission by 28.9% to 1974.6%, most significantly in the $N_{160}$ treatment. However, there was no clear effect
of WT levels on $N_2O$ emissions. We observed a significantly positive and linear relationship between the cumulative $N_2O$
emissions and N application doses, and the slope and intercept of the linear correlation depended on the WT levels (Figure 3).
The linear results also showed that the 1 kg N ha$^{-1}$ addition increased the cumulative $N_2O$ emission by 0.0028, 0.0024 and



0.0023 g $N_2O$-N $m^{-2}$ $yr^{-1}$ under $WT_{-30}$, $WT_0$ and $WT_{10}$ conditions, respectively.
**3.4 Global warming potential**

During the two study years, the combined GWP from the cumulative $CH_4$ and $N_2O$ emissions varied within the range

11.17 g $CO_2$-eq $m^{-2}$ to 1113.86 g $CO_2$-eq $m^{-2}$ (Figure 2 and Table S1). Both the WT and N treatments and their combination
showed significant effects on GWP (Table 3). According to the GAM results (Figure 4), the higher WT level increased GWP,
the predicted curve following the order: $WT_{10} > WT_0 > WT_{-30}$. Similar to the cumulative $CH_4$ emissions, the GWP emission
showed a non-linear relationship with N deposition levels under $WT_0$ and $WT_{10}$ conditions (Figure 4), and the highest value
occurred in the $N_{20}$ treatment. Nevertheless, under $WT_{-30}$ conditions, the response of GWP to N deposition was positive and
linear and thus similar to the pattern of $N_2O$, with the highest value being recorded in the $N_{160}$ treatment. Meanwhile, the
contribution of cumulative $CH_4$ emissions to GWP was negative (-112% to -4%) in the $WT_{-30}$ treatment but much higher in
the $WT_0$ (3% to 97%) and $WT_{10}$ (35% to 97%) treatments, especially for the $N_{20}$ treatment (84% to 97%) (Table S1). The
contribution of cumulative $N_2O$ emissions to total GWP was highest in the low WT level ($WT_{-30}$) and high N deposition
treatments ($N_{80}$ and $N_{160}$).

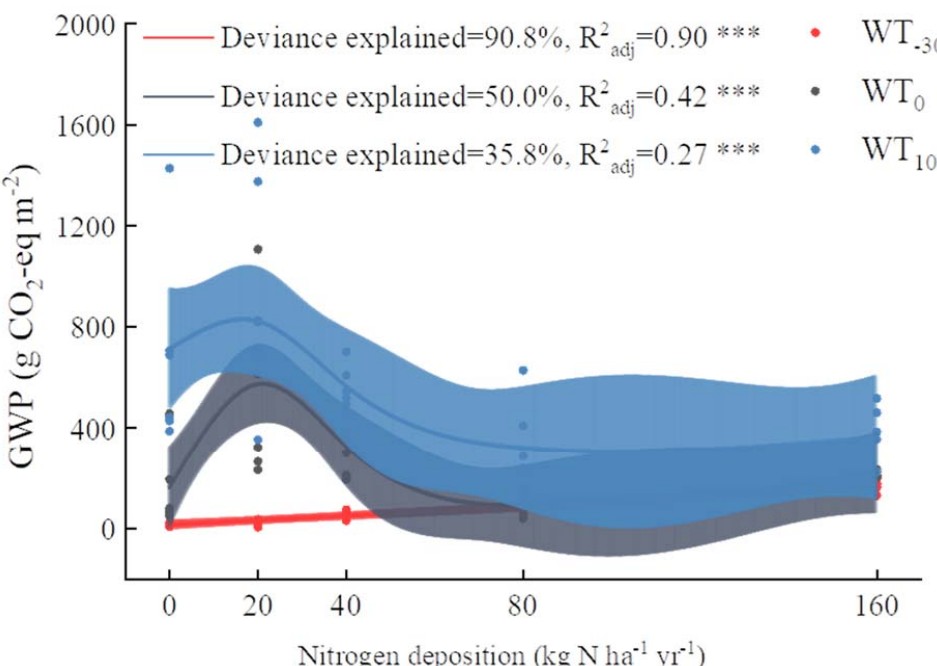

Figure 4. Effects of nitrogen deposition levels on the global warming potential (GWP) of greenhouse gas ($CH_4$ and $N_2O$)
budgets at three water table levels using generalized additive models. The statistical significance levels used were: * = $P <$



0.05 and > 0.01; \*\* = $P$ < 0.01 and > 0.001; \*\*\* $P$<0.001; NS = not significant ($P$ > 0.05). The shaded areas indicate 95%
confidence intervals.

**4. Discussion**

To better understand the sensitivity of alpine peatlands to climate change and anthropogenic disturbance, our mesocosm

experiment focused on the $CH_4$ and $N_2O$ emissions in response to multiple levels of N deposition at three WT levels in an
alpine peatland. Ours is one of few studies (Gao et al. 2014, Wang et al. 2017) exploring the interactive effects of WT levels
and N deposition on GHG emissions in alpine peatlands. Our main finding was that N deposition non-linearly affected the
cumulative growing season $CH_4$ emissions and linearly enhanced the $N_2O$ emission, the effect being highly dependent on the
WT levels. Our results partially supported hypothesis I – that N deposition would consistently enhance the $N_2O$ emissions and
that moderate N deposition would increase and high N deposition decrease the emissions (hypothesis I). This provides clear
evidence of the interactive effects of WT levels and N deposition on GHG emissions, supporting hypothesis II.
**4.1 Effects of WT and N deposition on $CH_4$ emission**

In the two-year mesocosm experiment, corresponding with the results of previous studies (Hoyos-Santillan et al. 2019,

Wang et al. 2017), WT levels had a significant positive effects on $CH_4$ emissions, and the higher WT levels generally increased
the SWC and $CH_4$ emissions. The extent of the $CH_4$ release from the Zoige peatland to the atmosphere depended on the balance
between the microbial processes of $CH_4$ production and oxidation. At the higher WT levels, the oxygen content in the surface
peat declined and the exposure of $CH_4$ production process to anaerobic conditions increased, elevating $CH_4$ emissions (Evans
et al. 2021, Hoyos-Santillan et al. 2019, Zhang et al. 2020).

The effects of N deposition on peatland $CH_4$ emissions reported in previous studies varied considerably with positive

(Juutinen et al. 2018), negative (Gao et al. 2014) or neutral (Wang et al. 2017) relationships reported. In the current study, a
moderate level of N deposition positively stimulated the $CH_4$ emissions, but subsequently the positive effect declined with
further N addition, despite the extremely low uptake or emission of $CH_4$ at low WT levels. This result is not unique as those
of Li et al. (2012), Lafuente et al. (2020) and Qu et al. (2021) all showed a similar non-linear pattern for grasslands, drylands
and steppe, respectively, although the aforementioned three regions acted as $CH_4$ sinks. The response of $CH_4$ emissions to N
deposition was depending on a threshold value of added N doses. Below this threshold, we observed positive effect – the
increasing N input alleviated of N constraints on microbial metabolism and, finally, increased $CH_4$ production (Currey et al.
2009, Deng et al. 2019). Above the threshold, the large amounts of available $NO_3^-$ led to negative and inhibitory effects on the
methanogenic activity due to the competition of $NO_3^-$-reducing bacteria with methanogens (Liu et al. 2020). In addition, $NO_3^-$



might be used as an alternative electron acceptor to $O_2$ in order to support the microbial $CH_4$ oxidation process, and thus the
produced $CH_4$ was more consumed by the methanotrophic microbes (Qu et al. 2021, Wang et al. 2017).
The interactive effects of WT levels and N deposition on the cumulative $CH_4$ emissions were distinct in our study (Table
3 and Figure 3). Thus, we found that the WT levels were more likely to determine the direction and magnitude of $CH_4$ emissions
from alpine peatlands than N deposition. The N deposition non-linearly affected the $CH_4$ emissions, but the optimal scenario
for $CH_4$ emissions was roughly ca. 20 kg N ha$^{-1}$ yr$^{-1}$, which could be slightly changed by the WT levels. This was not consistent
with the findings in previous studies showing no interactive effects of WT and N treatments on the $CH_4$ uptake or emissions
in the Qinghai-Tibetan Plateau (Wang et al. 2017, Wu et al. 2020). A possible explanation of this may be that WT levels may
significantly influence the positive or negative effects of N deposition on the microbial process of $CH_4$ production due to its
limitation on peat anaerobic conditions. It is likely that the manipulated N supply and aerobic conditions simultaneously
affected the microbial processes of $CH_4$ production and oxidation and thereby determined the level of $CH_4$ emissions. We also
speculate that the WT levels were associated with nutrient exploitation by microorganisms and, accordingly, that the higher
WT levels promoted diffusion of the added N in the water-filled soil pore, N thus becoming readily accessible in the microbial
process (Wang et al. 2017).
**4.2 Effects of WT and N deposition on $N_2O$ emission**
The mesocosms in the Zoige alpine peatland acted consistently as a $N_2O$ source in the two years, and we observed
significant positive and linear effects of N deposition on $N_2O$ emissions but no clear pattern of WT effects. This is not unique,
and Wang et al. (2017) reported that elevated WT levels under drained to inundated conditions had no effects on the $N_2O$ flux
in the alpine peatlands of the Tibetan Plateau. The present results show that the $N_2O$ emissions from alpine peatland were
likely primarily determined by N deposition rather than by WT levels. This is partially consistent with the study of (Gao et al.
2014), which also indicated the N addition (5.0 g N m$^{-2}$ yr$^{-1}$) significantly increased and the higher WT level slightly decreased
$N_2O$ emissions in the alpine peatlands of the Qinghai-Tibetan Plateau. The alpine peatlands are generally under pressure by N
deficits and highly sensitive to the climate change (Squeo et al. 2006). The N input via deposition could supply more N
substrate and activate the microbial process of $N_2O$ production. Our study showed that the N deposition did increase soil TN
(F = 4.49, $P$ = 0.002); however, this did not exhibit a linear relationship similar to $N_2O$ emissions with N deposition. This is
common, and some studies have even reported no effects of N addition on TN on the alpine steppe and grasslands of the
Tibetan Plateau, respectively (Qu et al. 2021, Zhao et al. 2017). The possible reason is WT- or N deposition-induced availability
of N substrate in soil peat, which is favourable for microbial $N_2O$ production (Cui et al. 2016, Yue et al. 2021, Zhu et al. 2020).
Both the WT levels and N deposition may affect the N availability in top layer soil and consequently regulate the denitrification
process (Han et al. 2019, Wang et al. 2017).



Studies on the interactive effects of WT levels and N deposition on $N_2O$ emissions in peatlands are rare and have
contradictory findings. For example, Wang et al. (2017) revealed no interactive effects of WT and N addition on $N_2O$ emissions
in the alpine peatlands of the Tibetan Plateau, while Gao et al. (2014) showed the N addition increased the $N_2O$ emissions;
albeit this was slightly inhibited by water addition. Our results confirmed the occurrence of an interactive effect of WT and N
deposition on $N_2O$ emissions, but it was neither synergistic nor antagonistic. N deposition had linear positive effects on $N_2O$
emission, and WT did not alter this linear relationship but slightly changed the slope and intercept. This finding is quite novel,
and we have not identified any similar results in the previous studies. Because of this, we speculate that the N supply was the
primary factor determining the $N_2O$ emissions in the N-limited Zoige alpine peatland, but this process was associated with the
WT levels via impacts on the efficiency of utilising N substrate for microbial $N_2O$ production.
The growing season (June to September) $N_2O$ emission responded to the 1 kg N $ha^{-1}$ $yr^{-1}$ deposition and increased by
0.0023-0.0028 g N $m^{-2}$ $yr^{-1}$ in the current study. This is slightly lower than the levels of previous studies, for instance that of
(Gong et al. 2019), who found that 1 kg annual N $ha^{-1}$ addition led to an increase of ca. 0.0076 g $N_2O$-N $m^{-2}$ $yr^{-1}$ during the
growing season (May to October) in a boreal peatland. This could be attributed to the relatively low air temperature at this
particular alpine peatland, which hampered the microbial $N_2O$ production. Furthermore, IPCC (2013) suggested that the
emission factor (the fraction of nitrogen added that is released as $N_2O$) is 1%, indicating that 1 kg annual N $ha^{-1}$ may increase
$N_2O$ emissions by 0.001 g N $m^{-2}$ $yr^{-1}$. The relatively higher emission factor for $N_2O$-N in our study was probably due to the
high dose of N addition.
**4.3 Effects of WT and N deposition on GWP**
The 100-year GWP was used as a tool to quantify the combined effect of $CH_4$ and $N_2O$ emissions and its response to N
deposition at different WT levels in the investigated Zoige alpine peatland. The response of GWP to the N deposition was
complex, and the largest increase in the GWP occurred at high WT levels (i.e. WT at soil surface or above) and ca. 20 kg N
$ha^{-1}$ $yr^{-1}$ deposition. This is partially consistent with the results of a previous study, which indicated that the threshold value of
N addition rates for the highest 100-year GWP ($CO_2$, $CH_4$ and $N_2O$) in the Qinghai-Tibetan Plateau was 20 kg N $ha^{-1}$ $yr^{-1}$ in
2013 (Qu et al. 2021). We also observed that the elevated WT levels increased the magnitude of total GWP in the region due
to the contribution of largely enhanced $CH_4$ emissions, this being supported by an earlier study in Qinghai-Tibetan Plateau by
Wang et al. (2017) indicating that the strength of the $CH_4$ source was the primary reason for the changed GWP in response to
the changed WT levels. Meanwhile, N deposition showed linear and non-linear effects on GWP depending on the WT levels.
The reason behind this was the main contributor to the total GWP changed from $CH_4$ to $N_2O$ along the elevated WT levels,
which led to a change in the response of total GWP to N addition from linear to non-linear. This result is quite novel, and
further studies should be conducted with focus on the long-term interactive effects of WT and N deposition on GWP from $CO_2$,





$CH_4$ and $N_2O$ combined.

**4.4 Implications for future GHG emissions in alpine peatlands**

The undisturbed peatlands are currently a weak carbon sink (~0.1 Pg C yr$^{-1}$) with a moderate source of methane (~0.03
Pg $CH_4$ yr$^{-1}$) and a very weak source of nitrous oxide (~0.00002 Pg $N_2O$-N yr$^{-1}$), but artificial drainage transitioned it into a
net C source (~0.1 Pg C yr$^{-1}$), with a 10% decrease in $CH_4$ emissions and 20-fold increase in $N_2O$ emissions (Frolking et al.
2011). In addition, a further study indicated that if we do not rehabilitate or restore the drained peatlands, they might constitute
12-41% of the GHG emissions budget by 2100 (Leifeld et al. 2019). Restoration of the drained peatlands is already regarded
as crucial to safeguard natural ecosystems and decrease GHG emissions (Laine et al. 2019), particularly to meet the
requirement of mitigation of GHG emissions in the Paris Agreements. Numerous studies have confirmed the relief of GHG
emissions (primarily mitigation of $CO_2$ but acceleration of $CH_4$) from peatlands due to the artificial restoration (Evans et al.
2021, Laine et al. 2019). However, accurate estimates of $CH_4$ and $N_2O$ emissions are still unavailable, in particularly associated
with the climate change.
Alpine or high-altitude peatlands constitute an exceptional group among the global peatlands (Le et al. 2020) and
traditionally, they are more fragile water bodies and sensitive to climate changes and human disturbances (Squeo et al. 2006).
The Zoige alpine peatland, the largest and highest wetland area in Qinghai-Tibetan Plateau, accounts for 6.2% of the SOC
storage in China and 1% in the world (Cao et al. 2017). Previous studies have already shown the Qinghai-Tibetan Plateau is
experiencing increasing N deposition (Zhu et al. 2020), drainage (Zhang et al. 2011) or ecological restoration (Xu et al. 2021).
However, the response of this vulnerable but large carbon pool to the increasing N deposition and fluctuating WT levels as
well as its contribution (particularly $CH_4$ and $N_2O$ emissions) to the future global GHG budgets is still uncertain.
Our study estimated the $CH_4$ and $N_2O$ emissions from an alpine peatland in response to the multi-level increasing N
deposition and fluctuating WT levels, and our results do not argue against full restoration or drainage of peatlands or the
reliability of the maybe too high level of N deposition. Our scenario of water table level (at soil surface or above) and N
deposition (ca. 20 kg N ha$^{-1}$ y$^{-1}$) effectively increased the $CH_4$ and $N_2O$ emissions from the studied Zoige alpine peatland.
Furthermore, additional N deposition enhanced the $N_2O$ emissions and slightly increased or even inhibited $CH_4$ emissions.
Considering the current N deposition of 1.08-17.81 kg N·ha$^{-1}$ in the Qinghai-Tibetan Plateau (Han et al. 2019) and of 21.1 kg
N ha$^{-1}$ in China (Liu et al. 2013), the future trend predicting a possible doubling or tripling of N deposition by the end of the
century (Lamarque 2005) as well as the restoration of drained peatlands to safeguard their function and to mitigate GHG
emissions, we need to pay attention to the probably increasing $CH_4$ emissions and the shift in the presently underestimated
$N_2O$ emissions from low to high levels. We believe that our results are strongly useful for predicting the GHG emissions from
alpine peatlands in response to the climate change and anthropogenic activities in the future.





**5. Conclusion**

The N-limited fragile alpine peatlands are highly sensitive to climate change and anthropogenic activities. Hence, we need to improve our understanding of the response of GHG emissions from alpine peatlands to the increasing nitrogen deposition and changing water table levels. Our results showed that $CH_4$ emissions were determined by N deposition, WT levels and their interactive effects. A modest input of N deposition and high WT levels both stimulated $CH_4$ emissions. $N_2O$ emissions were remarkably sensitive to N deposition, which consistently and linearly increased the $N_2O$ emissions, irrespective of WT levels. The N deposition supplied more nutrients and substrate for the GHG-related microbes, while the WT levels determined the soil aerobic conditions. Moreover, we speculated that the WT levels influenced the exploitation of nutrients for $CH_4$ and $N_2O$ production. The highest GWP was observed at high WT levels and ca. 20 kg N $ha^{-1}$ $yr^{-1}$ deposition. The current annual N deposition in the Qinghai-Tibetan Plateau (1.08-17.81 kg N·$ha^{-1}$) is not as high as in China as a whole (21.1 kg N·$ha^{-1}$). However, the projected increasing N deposition suggests that the GHG emissions from alpine peatlands have not yet peaked, and there is therefore a risk for higher $CH_4$ and $N_2O$ emissions in the future.

**Data availability**

All data are available from the corresponding author by request.

**Author contributions**

**Wantong Zhang:** Conceptualization, Data curation, Software, Writing - original draft, Writing - review & editing, Validation, Formal analysis. **Zhengyi Hu:** Conceptualization, Supervision. **Joachim Audet:** Data curation, Supervision, Writing – review & editing. **Thomas A. Davidson:** Supervision, Data curation, Writing – review & editing. **Enze Kang:** Investigation. **Xiaoming Kang:** Investigation, Project administration. **Yong Li:** Investigation. **Xiaodong Zhang:** Investigation, Project administration. **Jinzhi Wang:** Investigation, Conceptualization, Project administration, Writing - review & editing..

**Competing interests**

The contact author has declared that neither they nor their co-author has any competing interests.

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
