# Peer review of "Figure S1. Variations of daily air temperature and monthly precipitation during the growing seasons in"

_Biogeosciences, 2022_

## Author Comment (AC1)

Comments 1:

The manuscript is working on an important topic with a clear objective: to examine the response of non-$CO_2$ emissions (i.e., $CH_4$ and $N_2O$) to the manipulation of water table and N deposition in an alpine peatland ecosystem. Unfortunately, several major concerns from methodologies and discussions in the current draft make it unacceptable for publication by Biogeosciences.

major comments:

1. the sampling frequency in the year 2018 is too low to capture the temporal variation of the gas fluxes (only five sampling events were conducted over five months). Therefore, the cumulative emissions calculated contain high uncertainty.

Reply: Thanks for the comments. At the beginning of the experiment in 2018, we planned to measure GHG fluxes once a month, and this was confirmed by the previous study listed below, which used monthly GHG flux to assess the temporal variability and cumulative emissions. We agree that monthly sampling is not optimal for investigating temporal variations in 2018, and so we measured the GHG emissions frequently in 2019. We hope that the 2018 data could be better supported by the more frequent 2019 data. Furthermore, our main goal was to compare the different treatments and not to assess temporal variations. Still we agree that the high uncertainty connected to the cumulated fluxes should be better acknowledged and we have added this in the discussion. In Line 245-247 as follows:

R. Cao, Y. Chen, X. Wu, Q. Zhou, Sun S (2018) The effect of drainage on $CO_2$, $CH_4$ and $N_2O$ emissions in the Zoige peatland: a 40-month in situ study. Mires and Peat 21:1-15

It is not known whether the low frequency of GHG sampling in 2018 could cause uncertainties in the cumulative $CH_4$ emissions in 2018, but this was better supported by the more frequent events in 2019.

2. the second hypothesis points to the altered "efficiency of utilising nutrients for $CH_4$ and $N_2O$ production" by regulation of redox conditions through water table manipulation. However, no data in the current study can support such a hypothesis.

Reply: In fact, we did lack of enough data to fully support the hypothesis about utilizing nutrients for $CH_4$ and $N_2O$ production by regulation of redox conditions through water table manipulation. And so, we focus on the following hypotheses as follows in L72-74:

The effects of N deposition on $CH_4$ and $N_2O$ emissions would be associated with WT levels, with high $CH_4$ and $N_2O$ emissions at high WT levels.

3. the global warming potential (GWP) in the current study simply sums up the non-$CO_2$ emissions (based on their radiative forcing). Without including net ecosystem $CO_2$ exchange, or change of SOC stock, critical limitations exist in the significance of this work (the effects of treatments (water table and N deposition) on the GHG budget of the studied ecosystem). The elevated $CH_4$ emissions under a higher water table could be offset sufficiently by the depressed SOC decomposition, leaving the net effect unclear.

Reply: Thanks for your comments. We agree that our study could not show the GWP budget without data from $CO_2$ exchange or SOC stock, and hence, we deleted all the results about the GWP in the manuscript. We are now trying to focus on the non-$CO_2$ GHG emissions, to see the response of $CH_4$ and $N_2O$ emissions in the peatland to the increased N deposition levels at different water table levels.

4. discussions are generally shallow, and some parts are inappropriate. For example, many discussions are more like introductions instead of discussions (e.g., L276-277;

L316-319; L348-365, etc.). Section 4.4 is simply not implications, but background information that fits appropriately in the introduction, except for a few lines in the last paragraph. Discussions on denitrification and microbial $N_2O$ production (L309-315) are weak as soil TN is the only N measured. Discussions state some findings are "quite novel" (L321-322; L344-346) but fail to justify them (what is the implication and the potential contribution/influence if these are considered novel findings?)

Reply: We have carefully revised the discussion based on the reviewer comments. We have thoroughly revised the discussion of the manuscript, including the Section 4.4 implications (now Section 4.3). We carefully revised the parts about denitrification and microbial $N_2O$ production as follows in L291-293. We have already deleted the "quite novel" parts and revised the implications in Section 4.3.

The N deposition increased soil TN (F = 4.49, *P* = 0.002) in our study and is likely to supply more N substrate ($NH_4^+$ and $NO_3^-$) in soil (Zhu et al., 2020). The consequently increased N substrate could potentially activate the microbial process of $N_2O$ production and increase $N_2O$ emissions (Yue et al., 2021).

specific comments:

1. the units of the cumulative emissions (i.e., the main result) are confusing. Why are they "g C/N m-2 yr-1"? Based on the equation provided (L162), they should be "g C/N m-2" and calculated by integration over the growing season. Did the authors extrapolate the calculation to the non-growing seasons?

Reply: Thanks for helping us revising these mistakes, we revised all the units for cumulative emissions as g C/N $m^{-2}$ instead of g C/N $m^{-2}$ $yr^{-1}$ in the manuscript. We did not extrapolate the calculation to the non-growing seasons.

2. related to the question above, what happens to the non-growing season? any gas sampling was conducted from the mesocosm? Due to the low temperature, probably soils are frozen and thus the microbial activities are low, but the authors are recommended to include the explanation in the methodology and justify (with proper references) that emissions from growing seasons heavily dominated the gas fluxes.

Reply: We did not conduct any sampling in the non-growing season during the two years. As you mentioned, the low temperature and microbial activity in soil implied that the non-growing seasons GHG emissions have only a minor contribution to the yearly budget, which was also confirmed in the previous study, listed below. We also justified in the methodology about the GHG sampling only in the growing seasons in L113-116.

Reference: Peng H., Guo Q., Ding H. et al. Multi-scale temporal variation in methane emission from an alpine peatland on the Eastern Qinghai-Tibetan Plateau and associated environmental controls[J], Agricultural and Forest Meteorology, 2019, 276-277.

Justified part: The yearly budget of GHG emissions in the Zoige peatland was dominated by the growing season GHG emissions (Peng et al., 2019), therefore, we measured the $CH_4$ and $N_2O$ fluxes with the sampling events of 1-3 times per month during the growing seasons in 2018 and 2019 in our study. In total, 16 sampling occasions of individual fluxes were recorded for $CH_4$ and $N_2O$.

3. another unit issue for GWP. if cumulative emissions have the unit of "g C/N m-2 yr-1", why GWP ended up with "g $CO_2$-eq m-2" based on the equation provided (L171)? shouldn't it be "g $CO_2$-eq m-2 yr-1"?

Reply: We deleted all parts related to GWP, while focusing on the non-$CO_2$ emissions in the peatland.

4. the experimental design on the levels of N deposition includes an unrealistically high dose (i.e., 160 kg N ha$^{-1}$ y$^{-1}$). It is fine to examine the relationships, but proper efforts should be made to justify such a design in the discussion.

Reply: In consideration of the possible non-linear relationship between the GHG emissions and N deposition levels, multi levels of N depositions are essential in the experimental design. We did discuss about the levels of N deposition in the experiment, and probably the level of 0-40 kg N ha$^{-1}$ y$^{-1}$ was high enough to stimulate the near future N deposition. The reason for designing such a high dose (i.e., 80 or 160 kg N ha$^{-1}$ y$^{-1}$) is that we want to consider the possible N input from fertilizers or livestock excreta. we believe the additional results generated from the high-level N deposition could best support the conclusion of relationships between the GHG emissions and N deposition levels from 0-40 kg N ha$^{-1}$ y$^{-1}$, making the linear or non-linear fitting more reliable. Moreover, this is not the first design of such a high level of N deposition in the Qinghai-Tibetan Plateau (listed below), and we briefly added one sentence (listed below) in the methodology to support the design of such a high dose of N deposition in L106-108.

The added sentence: The three lowest levels ($N_0$, $N_{20}$ and $N_{40}$) are covering the gradient of current and near-future deposition levels while the two highest levels ($N_{80}$ and $N_{160}$) represent levels of N-enrichment resulting from extreme deposition possibly levels possibly combined with fertilization.

The new Referece: Qu S., Xu R., Yu J. et al. Nitrogen deposition accelerates greenhouse gas emissions at an alpine steppe site on the Tibetan Plateau[J], Science of the Total Environment, 2021, 765: 144277.

5. for the calculation of cumulative emission, the authors can simply describe it like "linear interpolation between sampling events using the trapezoidal rule" instead of providing the equation and explanation for the notations (L161-166). instead, the authors are recommended to provide equations for calculating the gas fluxes rather than

simply saying "calculated by the slopes of linear regression between gas concentrations" (is it corrected with temperature? atmospheric pressure?)

Reply: We revised the description about the calculation of GHG flux in L161-164, and revised the description for cumulative emissions in L128-134.

some minor corrections:

1. L77: highlevel -> high-level N deposition

Reply: We revised it in L72 as follows.

a slight increase in N deposition might stimulate both $CH_4$ and $N_2O$ emissions, but a high-level N deposition would inhibit $CH_4$ emissions while $N_2O$ emissions would reach a threshold.

2. L78: "aerobic conditions"? are the authors trying to mention redox conditions? similar expressions occur in several parts of the remaining text, consider rephrasing (e.g., L295, L383).

Reply: Yes, we meant altered redox conditions, and we revised the aerobic conditions as redox conditions in the manuscript.

3. descriptions of the mesocosm design and treatment manipulation are not very clear, the authors are recommended to include a supplementary figure for a clear illustration. in particular, the definition of $WT_0$ can be confusing (i.e., soil-water interface; L101), is it simply "the soil surface"?

Reply: We are sorry this phrase led to some confusion. In our study, the WT$_0$ means that the water level is just at the soil surface. We reformulated this explanation in the manuscript, and we added a supplementary figure to better illustrate the design as follows.

[Figure]

4. L158: despite -> regardless of. Also, the description of how the GAM is applied could be oversimplified. Ask this question may help the authors to improve the description: does the current description sufficient for peer researchers to reproduce the analysis?

Reply: We now revised the details about the application of GAM in the R listed below in L157-159. We hope this description could be clear enough to reproduce to the analysis.

Via the R package "mgcv" (Wood, 2017), we used method "gam" to perform the GAM analysis and method "predict.gam" to see the response value of GHG emissions along the N deposition gradient from 0 to 160 kg N ha$^{-1}$ yr$^{-1}$.

5. L166-168. difficult to follow. how can the heterogeneity be reduced?

Reply: We now rephrased the description about this part listed below in L161-164.

The cumulative GHG emissions in the growing seasons of each year were calculated by linear interpolation between sampling events using the trapezoidal rule (Goldberg et al., 2010). In addition to the cumulative GHG emissions between the first and the last sampling event, the GHG emissions from 1st June to the first sampling and from the last sampling to 30th September were taken into consideration.

6. L173-174. reference missed.

Reply: Thanks, but we deleted all the parts about GWP including L173-174.

7. L175. "by applying the statistic R software" -> "using R"

Reply: we now revised it in L165 as follows:

Statistical analysis was carried out using R (version 3.4.3) (R Development Core Team, 2017).

8. L180. SWC has been abbreviated in L146.

Reply: Thanks, we already revised it in L148 as follows:

SWC was determined by using a TDR300 moisture meter (Spectrum Technologies Inc., Plainfield, Illinois, USA).

9. L205-206: "the highest value occurring"-> "with the highest value observed"

Reply: we now revised it in L194.

The response of the cumulative $CH_4$ emissions to N deposition was non-linear under $WT_0$ and $WT_{10}$ conditions (Figure 3), with the highest value observed in the $N_{20}$ treatment.

10. L245: "combination" -> "interaction"

Reply: We deleted the GWP part including this.

11. L267: add "CH4" before "emissions"

Reply: we revised it in L237 listed below.

This result partially supported hypothesis I – that N deposition would consistently enhance $N_2O$ emissions and that increasing N deposition would increase $CH_4$ emissions until a threshold is reached (hypothesis I).

12. L273-274: needs rephrasing. Note that the study did not measure oxygen content, and therefore the expression like "oxygen content declined" is not appropriate. Consider: "With higher WT levels, SWC increased and likely formed more anaerobic conditions conducive to $CH_4$ production, leading to elevated $CH_4$ emissions (references)."

Reply: Thanks, and we revised the sentence listed below in L249-251.

With higher WT levels, SWC increased and likely formed more anaerobic conditions conducive to $CH_4$ production, leading to elevated $CH_4$ emissions (Evans et al., 2021, Hoyos-Santillan et al., 2019, Zhang et al., 2020).

13. L276: add comma after "considerably"

Reply: We revised this whole part and deleted the "considerably".

14. L293-295: difficult to follow.

Reply: We rephrased the whole part to make it clear and readable listed below in L267-269.

We speculate that the WT levels were associated with nutrient exploitation by microorganisms and, accordingly, that the higher WT levels promoted diffusion of the added N in the water-filled soil pore, N thus becoming readily accessible in the microbial process (Wang et al., 2017).

15. L305: reference format: Gao et al. (2014).

Reply: We revised this part in Discussion and deleted it.

16. L343: from $CH_4$ to $N_2O$ -> from $N_2O$ to $CH_4$

Reply: We deleted the whole part of GWP including "from $CH_4$ to $N_2O$".

17. L369: increased -> decreased

Reply: We revised the the whole part of Section 4.3-Implications for future GHG emissions in alpine peatlands, including "increased".

---

## Author Comment (AC2)

Comments 2:

Wetland is an important source of $CH_4$ and $N_2O$. Global change especially changes in precipitation and N deposition could have greatly effect on them. However, how do they affect fluxes of $CH_4$ and $N_2O$ is still unclear in wetland on the Qinghai-Tibetan Plateau. This manuscript focused on the effects of nitrogen deposition on $CH_4$ and $N_2O$ emissions under three water table levels in the Zoige alpine peatland. Thus, it is an important and interesting topic. However, there are still minor flaws that should be revised prior possible publication by this journal.

1. The present results are relying on the five levels of nitrogen deposition, but some levels (such as 160 kg N ha$^{-1}$ yr$^{-1}$) are extremely higher compared to the local nitrogen deposition (1.08-17.81 kg N ha$^{-1}$ yr$^{-1}$), could authors explain why to design the treatments?

Reply: Thanks for the comments. Before initiating the experiment, we did consider about the design of N deposition levels. However, we still kept the extremely high-level N deposition is because we want to consider the excessive and possible N input from fertilizers or livestock excreta and we also believed this high-level N deposition would not hamper us to draw a conclusion, but supporting the results even further. In fact, our main results about the relationships between the GHG emissions and N deposition are based on the N deposition levels of 0-40 kg N ha$^{-1}$ yr$^{-1}$ or 0-80 kg N ha$^{-1}$ yr$^{-1}$, and the higher level is just used to examine and confirm the relationship. Moreover, this is not the first design of such a high level of N deposition in the Qinghai-Tibetan Plateau (listed below), and we briefly added one sentence (listed below) in the methodology to support the design of such a high dose of N deposition in L106-108.

The added sentence: The three lowest levels ($N_0$, $N_{20}$ and $N_{40}$) are covering the gradient of current and near-future deposition levels while the two highest levels ($N_{80}$ and $N_{160}$) represent levels of N-enrichment resulting from extreme deposition possibly levels possibly combined with fertilization.

The new Referece: Qu S., Xu R., Yu J. et al. Nitrogen deposition accelerates greenhouse gas emissions at an alpine steppe site on the Tibetan Plateau[J], Science of the Total Environment, 2021, 765: 144277.

2. Authors conducted a two-year mesocosm experiment, how about the variability of soil properties and GHG emissions within the two years. Suggest you to compare the differences of SOC, TN or GHG emissions between 2018 and 2019.

Reply: Thanks for the suggestion. We did check for the yearly differences of SOC, TN, CH₄ and N₂O emissions between 2018 and 2019, and the figure is as follows. Unfortunately, we haven't found any clear patterns about them, and so we did not put this figure in the manuscript.

[Figure]

3. It is better to revise the second hypothesis to "The effects of N deposition on CH₄ and N₂O emissions would be associated with WT levels" in lines 77-79.

Reply: We rephrased the second hypothesis listed below in L72-74.

The effects of N deposition on CH₄ and N₂O emissions would be associated with WT levels, with higher CH₄ and N₂O emissions at high WT levels.

4. Discussion should be improved, some parts are just a repeat from the Introduction.

Reply:   We have carefully and thoroughly revised the whole part of discussion.

5. English in the manuscript should be improved.

Reply: We have revised the language with help of a native speaker

Specific mistakes:

(1) delete "1% in IPCC" in the Abstract.

Reply: we deleted it.

(2) the sentence of "the large carbon pool is nitrogen deficient and is recognized …." in lines 32-33 is hard to understand and need to be rewritten.

Reply: We rephrased the sentence listed below in L30-31.

Traditionally, this nitrogen-limited ecosystem is recognized as major $CH_4$ sources and weak $N_2O$ sources (Frolking et al., 2011).

Frolking S., Talbot J., Jones M. C., Treat C. C., Kauffman J. B., Tuittila E.-S. , Roulet N.: Peatlands in the Earth's 21st century climate system, Environ. Rev., 19, 371-396. https://doi.org/10.1139/a11-014, 2011.

(2) Delete "(mean ± SE) (n=3)" in the title of table 1.

Reply: Thanks for the comments, we deleted them in the title, and put them in the table foot in L180 as follows.

Each value represents mean ± SE (n=3). SWC, soil water content; SOC, soil organic carbon; TN, total nitrogen.

(4) line 90: July should be revised to June.

Reply: We revised it in L109 as follows:

The annual added N doses were further divided into four portions and applied at the beginning of every month from June to September in 2018 and 2019.

(5) line 213: the name of Figure 1 should be changed, it is hard to see the response of GHG flux to nitrogen deposition.

Reply: we revised the name of figure 1 in L201-202 listed below.

Figure 1. Temporal variation of growing-season $CH_4$ and $N_2O$ fluxes under five levels of nitrogen deposition (0, 20, 40, 80 and 160 kg N $ha^{-1}$ $yr^{-1}$) and three water table levels in 2018 and 2019. Error bars represent the SE (n=3).

(6) Line 217, "During the rowing seasons", rowing should be revised to growing.

Reply: We revised it in L205.

Figure 2. Effects of nitrogen deposition levels on cumulative $CH_4$ and $N_2O$ emissions at three water table levels during the growing seasons in 2018 and 2019. Error bars represent the SE (n=3).

(7)Line 274: "the exposure of $CH_4$ production process to anaerobic conditions increased" might to be changed to "$CH_4$ production under anaerobic conditions was increased".

Reply: We rephrase it in another way in L249-251.

With higher WT levels, SWC increased and likely formed more anaerobic conditions conducive to $CH_4$ production, leading to elevated $CH_4$ emissions (Evans et al., 2021, Hoyos-Santillan et al., 2019, Zhang et al., 2020).

Evans C. D., Peacock M., Baird A. J., Artz R. R. E., Burden A., Callaghan N., Chapman P. J., Cooper H. M., Coyle M., Craig E., Cumming A., Dixon S., Gauci V., Grayson R. P., Helfter C., Heppell C. M., Holden J., Jones D. L., Kaduk J., Levy P., Matthews R., McNamara N. P., Misselbrook T., Oakley S., Page S., Rayment M., Ridley L. M., Stanley K. M., Williamson J. L., Worrall F. , Morrison R.: Overriding water table control on managed peatland greenhouse gas emissions, Nature., https://doi.org/10.1038/s41586-021-03523-1, 2021.

Hoyos-Santillan J., Lomax B. H., Large D., Turner B. L., Lopez O. R., Boom A., Sepulveda-Jauregui A. , Sjögersten S.: Evaluation of vegetation communities, water table, and peat composition as drivers of greenhouse gas emissions in lowland tropical peatlands, Sci. Total Environ., 688, 1193-1204. https://doi.org/10.1016/j.scitotenv.2019.06.366, 2019.

Zhang W., Wang J., Hu Z., Li Y., Yan Z., Zhang X., Wu H., Yan L., Zhang K. , Kang X.: The Primary Drivers of Greenhouse Gas Emissions Along the Water Table Gradient in the Zoige Alpine Peatland, Water Air Soil Poll., 231, 5. https://doi.org/10.1007/s11270-020-04605-y, 2020.

(8) Figure S1, the precipitations from the peatland in June, August and September of 2019 were extremely high, reaching more than 2500 mm in one month. You should scrutinize the raw data.

Reply: We recheck the original data of the precipitations from the Zoige peatland in 2019, and we found a mistake in calculating the monthly precipitation generated from the daily precipitations. We now revised it and attached the new figure S1.

[Figure]

(9) line304: "show" should be revised to "showed".

Reply: We revised this whole part including L304.

(10) line 305-306: "…the study of (Gao et al. 2014)" should be revised to "…the study of Gao et al. (2014)".

Reply: We revised this whole part and deleted the original content in L305-306.

(11) line 306-307: revise the whole sentence to "which indicated that $N_2O$ emissions was significantly increased by N addition (5.0 g N $m^{-2}$ $yr^{-1}$) and slightly decreased in the higher WT level in the alpine peatlands of the Qinghai-Tibetan Plateau."

Reply: We revised this whole part and delete the original sentences in L306-307.

(12) line 313: "soil peat" should be revised to "soil".

Reply: We revised it in L292 as follows:

The N deposition increased soil TN (F = 4.49, $P$ = 0.002) in our study and is likely to supply more N substrate ($NH_4^+$ and $NO_3^-$) in soil (Zhu et al., 2020).

Zhu J., Chen Z., Wang Q., Xu L., He N., Jia Y., Zhang Q. , Yu G.: Potential transition in the effects of atmospheric nitrogen deposition in China, Environ. Pollut., 258, 113739. https://doi.org/10.1016/j.envpol.2019.113739, 2020.

(13) line 327: (Gong et al. 2019) should be revised to Gong et al. (2019).

Reply: We revised it in another way in L311-312, from "This is slightly lower than the levels of previous studies, for instance that of (Gong et al., 2019)" to "This is slightly lower than the levels from previous studies (Gong et al., 2019)".

Gong Y., Wu J., Vogt J. , Le T. B.: Warming reduces the increase in $N_2O$ emission under nitrogen fertilization in a boreal peatland, Sci. Total Environ., 664, 72-78. https://doi.org/10.1016/j.scitotenv.2019.02.012, 2019.

---

## Author Response (AR3)

**Comments from editor**:

Dear Authors, I am overall satisfied with the new version of the manuscript. You addressed all the reviewers' comments and also mines. I am pleased to endorse publication upon correction of some minor issues as listed below.

Please make the appropriate changes and upload a new version of the manuscript for publication.

L.24 replace "GHG" with "$CH_4$ and $N_2O$"

Reply: We revised it.

L.72 The first question is unclear. What do you mean "positive effects"? in your introduction you do not explain that N deposition has positive effect. Please rephrase to something like: "do increasing rates of N deposition stimulate $CH_4$ and $N_2O$ emissions?"

Reply: Thanks for the nice suggestion, and we revised it from "with the N deposition consistently increasing, do the positive effects of N deposition on $CH_4$ and $N_2O$ emissions persist?" to "do increasing rates of N deposition consistently stimulate $CH_4$ and $N_2O$ emissions?".

L.246 What do you mean by "but not reaching a significant level". please rephrase this sentence

Reply: We are sorry for this led to some confusion. We meant that the Song et al. (2013) found a similar pattern of non-linear relationship between N addition (0-240 kg N $ha^{-1}$ $yr^{-1}$) and $CH_4$ fluxes in a wetland, but unfortunately, they did not conduct any statistical analysis regarding this non-linear relationship. The authors only conducted the ANOVA to explore the overall effects of N addition on $CH_4$ emissions, which is not significant. Therefore, we rephrased it from "Song et al. (2013) reported that the N addition (0-240 kg N $ha^{-1}$ $yr^{-1}$) showed non-linear positive effects on $CH_4$ fluxes in a wetland with the highest $CH_4$ flux at 60 kg $ha^{-1}$ $yr^{-1}$ N addition, but not reaching a significant level." to "Song et al. (2013) found a similar pattern of non-linear effects of N addition (0-240 kg N $ha^{-1}$ $yr^{-1}$) on $CH_4$ fluxes in a wetland with the highest $CH_4$

flux observed at 60 kg ha$^{-1}$ yr$^{-1}$ N addition, but unfortunately, this N effect was not significant.".

L.252 Change "likely led" to "might have led"

Reply: We revised it.

L.256 Change "the previous study" with "a previous study"

Reply: We revised it.

L.259 Change "as far as we know" with "to the best of our knowledge"

Reply: We revised it.

**Comments from editor**:

Dear Authors, Thanks for submitting your article. As you have seen from the comments of the two referees, they point out numerous issues in which the manuscript needs revision before I can consider acceptance (especially reviewer number 1). I think that overall the study is still worth to be published and the comments are mainly referred on how authors interpret and use their results, thus I believe that the authors can address all of them. Please send a new version of the manuscript with the corrections implemented (following the reviewer's comments), upon which I can make a decision on whether accepting or not.

Reply: Thanks for the nice comments on our manuscript. We carefully revised the manuscript as you suggested as below. We think that the manuscript has been greatly improved after revision and hope it will be considered for publication.

Please pay special attention to the list below, which contains the major corrections needed and also some additional comments from me.

1. There are several limitations in this study. Please add a paragraph in your discussion called: "Study limitations" just before the conclusion. In this you should add an explanation of what your low sampling frequency in 2018 (and I would say that also 2019 is not that a high frequency) could implicate for cumulative calculations. As reviewer 1 highlight the sampling frequency is too low in 2018. This has nothing to do with temporal variability, but the fact that you cannot extrapolate precise measure of cumulative and neither you can compare treatments in that sense. You should also add that you did not measure other carbon related parameters (e.g. $CO_2$ or soil C) and thus the final relevance of your measurements on the contribution to feedback to climate change is limited. Finally, you should highlight that you only measure in the growing season, and while the non-growing season displays low emission could still modify the results you observed.

Reply: We combined "Study limitations" with Section 4.3, which is now renamed as "Implications and limitations". In the second paragraph of Section 4.3, we are

discussing the three main limitations in our study: 1. the extremely high level of N depositions in our study; 2. the absent/limited measurements for ecosystem $CO_2$ exchange and wintertime GHG emissions; 3. the low frequency of GHG sampling in 2018. Please see L317-328. Additionally, in the Discussion part, we compare our cumulative GHG emissions with other studies conducted in the peatland of the Qinghai-Tibet Plateau, just to make sure the generated conclusion in our study is reliable.

2. Nitrogen deposition rates: as highlighted by the reviewer 2 that deposition rates are much higher than reality. I do understand that adding high values will help establish a linear correlation. Please add this in your explanation and revise your answer to reviewer 2 (there are some typos in your current answer).

Reply: Thanks for the comment. The design for the present N deposition rates had already been observed in a previous study (Qu et al., 2021).

Qu S., Xu R., Yu J., Li F., Wei D. , Borjigidai A.: Nitrogen deposition accelerates greenhouse gas emissions at an alpine steppe site on the Tibetan Plateau, Sci. Total Environ., 765, 144277. https://doi.org/10.1016/j.scitotenv.2020.144277, 2021.

And, we have elaborated the reasons for such a design in L105-107, listed below:

The three lowest levels ($N_0$, $N_{20}$ and $N_{40}$) are covering the gradient of current and near-future deposition levels while the two highest levels ($N_{80}$ and $N_{160}$) represent levels of N-enrichment resulting from extreme deposition levels possibly combined with N input from fertilization or livestock excreta.

Also, we explained the uncertainties of the design related to the present results in the Discussion part in L317-320 and listed below.

It should be noted that some levels of N deposition (80 or 160 kg N ha$^{-1}$ yr$^{-1}$) in our study were much higher than the local N deposition (1.08-17.81 kg N ha$^{-1}$ yr$^{-1}$). This should not affect our general conclusion, because the non-linear and linear effects of N deposition on $CH_4$ and $N_2O$ emissions (respectively) were primarily dependent on the low levels of N deposition (0-40 kg N ha$^{-1}$ yr$^{-1}$), and the higher levels did not alter the relationship pattern.

3. Hypotheses: both reviewers point out some issue in one of the hypothesis, which is better in the revised form. However, it is still unclear how the authors concluded the current hypothesis, which do not seem properly support or explained with the offered background in the introduction. Please either make sure that the hypotheses are properly justified or remove the hypotheses and only leave a set of questions.

Reply: Thanks for the comments. We deleted the hypotheses and leave a set of questions in L71-73 listed below:

In this study, we aim to addres the following two questions: i) with the N deposition consistently increasing, do the positive effects of N deposition on $CH_4$ and $N_2O$ emissions persist? ii) if there is interaction between N depostion and WT level, how do they combine to influence $CH_4$ and $N_2O$ emissions in the alpine peatland ?

4. Discussion: please reduce the overall extent of the discussion and focus on a synthetic discussion of your results limiting speculations. Especially reviewer 1 points out several things to improve the discussion section. The section about implications should be seriously reduced. It could be combined with the section "study limitation " that I indicated in point 1, so that readers can get a good sense on to what extent the implications of your study can be used for management or calculation of future GHG rates (this is not to undermine your study but just to realistically discuss your results)

Reply: Thanks for the suggestions. We have carefully and thoroughly revised the Discussion, and combined the original section 4.3 with "study limitations", which is now specified as "Implications and limitations". We elaborated the implications and limitations of our study in a more concise and straightforward way.

Comments 1:

The manuscript is working on an important topic with a clear objective: to examine the response of non-$CO_2$ emissions (i.e., $CH_4$ and $N_2O$) to the manipulation of water table and N deposition in an alpine peatland ecosystem. Unfortunately, several major concerns from methodologies and discussions in the current draft make it unacceptable for publication by Biogeosciences.

major comments:

1. the sampling frequency in the year 2018 is too low to capture the temporal variation of the gas fluxes (only five sampling events were conducted over five months). Therefore, the cumulative emissions calculated contain high uncertainty.

Reply: Thanks for the comments. At the beginning of the experiment in 2018, we planned to measure GHG fluxes once a month, and this was confirmed by a previous study listed below, which used monthly GHG flux to assess the temporal variability and cumulative emissions.

Cao R., Chen Y., Wu X., Zhou Q., Sun S.: The effect of drainage on $CO_2$, $CH_4$ and $N_2O$ emissions in the Zoige peatland: a 40-month in situ study. Mires Peat, 21, 1-15. https://doi.org/10.19189/MaP.2017.OMB.292, 2018.

We agree that monthly sampling is not optimal for investigating temporal variations in 2018, and so we measured the GHG emissions frequently in 2019. Moreover, we compared our results with the previous studies conducted in the peatland of the Qinghai-Tibet Plateau in L233-235 and in L267-269.

The cumulative $CH_4$ emissions from the Zoige alpine peatland in our study ranged from -0.35 to 29.26 g $CH_4$-C m$^{-2}$ across the two years, which is in the same order of magnitude as the cumulative $CH_4$ emissions (25.4-29.6 g $CH_4$-C m$^{-2}$) from an alpine wetland of the Qinghai-Tibetan Plateau in a previous study (Wang et al., 2017).

The cumulative $N_2O$ emission from the Zoige peatland in our study was relatively higher than that in a previous report (0.08-0.2 g m$^{-2}$), which focused on the drainage or lower water table level (Cao et al., 2018).

We also revised the section 4.3 as "Implications and limitations", in which we discussed the uncertainties due to low frequency of GHG sampling in 2018 related to the present results. Still we agree that the high uncertainty should be better acknowledged and we suggested a further monitor of GHG fluxes to eliminate the possible uncertainties in L324-328 listed below.

In addition, the low frequency of GHG sampling in 2018 could cause uncertainties in the temporal variation and cumulative emissions of $CH_4$ and $N_2O$, and this might result in bias in the present result. The monthly measurements of GHG fluxes from peatlands have already been found in the previous study (Cao et al., 2018), and we also increased the sampling frequency in 2019 to better support the current conclusion. However, further monitoring of GHG fluxes from the peatland would still be required to eliminate the uncertainties.

2. the second hypothesis points to the altered "efficiency of utilising nutrients for $CH_4$ and $N_2O$ production" by regulation of redox conditions through water table manipulation. However, no data in the current study can support such a hypothesis.

Reply: The second hypothesis focused on the interactive effects of N deposition and WT level on $CH_4$ and $N_2O$ emissions, but we did lack of enough data to fully support. Therefore, we deleted the two hypothesis and leave the following two questions which could be supported by the present results, in L71-73, listed below.

In this study, we aim to address the following two questions: i) with the N deposition consistently increasing, do the positive effects of N deposition on $CH_4$ and $N_2O$ emissions persist? ii) if there is interaction between N depostion and WT level, how do they combine to influence $CH_4$ and $N_2O$ emissions in the alpine peatland?

3. the global warming potential (GWP) in the current study simply sums up the non-$CO_2$ emissions (based on their radiative forcing). Without including net ecosystem $CO_2$ exchange, or change of SOC stock, critical limitations exist in the significance of this work (the effects of treatments (water table and N deposition) on the GHG budget of the studied ecosystem). The elevated $CH_4$ emissions under a higher water table could

be offset sufficiently by the depressed SOC decomposition, leaving the net effect unclear.

Reply: Thanks for your comments. We agree that our study could not show the GWP budget without data from $CO_2$ exchange or SOC stock, and hence, we deleted all the results about the GWP in the manuscript. We are now trying to focus on the non-$CO_2$ GHG emissions, to see the response of $CH_4$ and $N_2O$ emissions in the peatland to the increased N deposition levels at different water table levels.

4. discussions are generally shallow, and some parts are inappropriate. For example, many discussions are more like introductions instead of discussions (e.g., L276-277; L316-319; L348-365, etc.). Section 4.4 is simply not implications, but background information that fits appropriately in the introduction, except for a few lines in the last paragraph. Discussions on denitrification and microbial $N_2O$ production (L309-315) are weak as soil TN is the only N measured. Discussions state some findings are "quite novel" (L321-322; L344-346) but fail to justify them (what is the implication and the potential contribution/influence if these are considered novel findings?)

Reply: We have carefully and thoroughly revised the Discussion part of the manuscript based on the reviewer comments. Particularly, the original Section 4.4 is now Section 4.3, which is renamed as "Implications and limitation". We reduced the whole extent of the Section 4.3, trying to elaborate the implications and limitations of our study in a more concise and straightforward way. Additionally, we carefully revised the parts about denitrification and microbial $N_2O$ production as follows in L277-279. We have already deleted the "quite novel" parts and put the novel implications of our study in Section 4.3.

The N deposition increased soil TN (F = 4.49, $P$ = 0.002) in our study and is likely to supply more N substrate ($NH_4^+$ and $NO_3^-$) in soil (Zhu et al., 2020). The consequently increased N substrate could potentially activate the microbial process of $N_2O$ production and increase $N_2O$ emissions (Yue et al., 2021).

specific comments:

1. the units of the cumulative emissions (i.e., the main result) are confusing. Why are they "g C/N m-2 yr-1"? Based on the equation provided (L162), they should be "g C/N m-2" and calculated by integration over the growing season. Did the authors extrapolate the calculation to the non-growing seasons?

Reply: Thanks for helping us revising these mistakes, we revised all the units for cumulative emissions as g C/N $m^{-2}$ instead of g C/N $m^{-2}$ $yr^{-1}$ in the manuscript. We did not extrapolate the calculation to the non-growing seasons.

2. related to the question above, what happens to the non-growing season? any gas sampling was conducted from the mesocosm? Due to the low temperature, probably soils are frozen and thus the microbial activities are low, but the authors are recommended to include the explanation in the methodology and justify (with proper references) that emissions from growing seasons heavily dominated the gas fluxes.

Reply: We did not conduct any sampling in the non-growing season during the two years. As you mentioned, the low temperature and microbial activity in soil implied that the non-growing seasons GHG emissions have only a minor contribution to the yearly budget, which was also confirmed in a previous study. We also justified the GHG sampling in the Section 4.3 "Implications and limitations" in L320-325.

Meanwhile, note must be made that we did not measure the net ecosystem $CO_2$ exchange and wintertime GHG fluxes, which might hamper estimating the annual carbon budget from GHG emissions and SOC change. However, our study focused on the growing-season non-$CO_2$ emissions from the peatland at different WT levels under the future scenarios of increasing N deposition, and also the non-growing-season GHG emissions had only a minor contribution to the yearly budget due to the low temperature and microbial activities (Peng et al., 2019).

Peng, H., Guo, Q., Ding, H., Hong, B., Zhu, Y., Hong, Y., et al.: Multi-scale temporal variation in methane

emission from an alpine peatland on the Eastern Qinghai-Tibetan Plateau and associated environmental controls. Agricultural and Forest Meteorology., 276-277, https://doi.org/10.1016/j.agrformet.2019.107616, 2019.

3. another unit issue for GWP. if cumulative emissions have the unit of "g C/N m-2 yr-1", why GWP ended up with "g $CO_2$-eq m-2" based on the equation provided (L171)? shouldn't it be "g $CO_2$-eq m-2 yr-1"?

Reply: We deleted all parts related to GWP, while focusing on the non-$CO_2$ emissions in the peatland.

4. the experimental design on the levels of N deposition includes an unrealistically high dose (i.e., 160 kg N $ha^{-1}$ $y^{-1}$). It is fine to examine the relationships, but proper efforts should be made to justify such a design in the discussion.

Reply: The reason for designing such a high dose (i.e., 80 or 160 kg N $ha^{-1}$ $y^{-1}$) is that we want to consider the possible non-linear relationship between GHG emissions and N deposition levels, as well as to include the possible N input from fertilizers or livestock excreta. we believe the additional results generated from the high-level N deposition could best support the conclusion of relationships between the GHG emissions and N deposition levels from 0-40 kg N $ha^{-1}$ $y^{-1}$, making the linear or non-linear fitting more reliable. We justified the design in the Section 2.3 in L105-107 listed below:

The three lowest levels ($N_0$, $N_{20}$ and $N_{40}$) cover the gradient of current and near-future deposition levels while the two highest levels ($N_{80}$ and $N_{160}$) represent levels of N-enrichment resulting from extreme deposition levels possibly combined with N input from fertilization or livestock excreta

We also discussed the uncertainties resulting from the design related to the present results in Section 4.3 in L317-320 listed below:

It should be noted that some levels of N deposition (80 or 160 kg N $ha^{-1}$ $yr^{-1}$) in our study were much higher than the local N deposition (1.08-17.81 kg N $ha^{-1}$ $yr^{-1}$). This should not affect our general

conclusion, because the non-linear and linear effects of N deposition on $CH_4$ and $N_2O$ emissions (respectively) were primarily dependent on the low levels of N deposition (0-40 kg N ha$^{-1}$ yr$^{-1}$), and the higher levels did not alter the relationship pattern.

5. for the calculation of cumulative emission, the authors can simply describe it like "linear interpolation between sampling events using the trapezoidal rule" instead of providing the equation and explanation for the notations (L161-166). instead, the authors are recommended to provide equations for calculating the gas fluxes rather than simply saying "calculated by the slopes of linear regression between gas concentrations" (is it corrected with temperature? atmospheric pressure?)

Reply: We revised the description about the calculation of GHG flux in L126-132,

The $CH_4$ and $N_2O$ fluxes were calculated as follows.

$$F = \frac{M}{V_0}\frac{P}{P_0}\frac{T}{T_0}\frac{dc}{dt}H$$

where dc/dt is the slope of the linear regression for the gas concentration gradient through time; M is the molecular mass of $CH_4$ or $N_2O$; P is the atmospheric pressure at the sampling site; T is the absolute temperature during sampling; $V_0$, $P_0$, and $T_0$ are the gas mole volume, atmospheric pressure, and absolute temperature under standard conditions, respectively; and H is the chamber height.

And also, we revised the description for cumulative emissions in L159-162.

The cumulative GHG emissions in the growing seasons of each year were calculated by linear interpolation between sampling events using the trapezoidal rule (Goldberg et al., 2010). In addition to the cumulative GHG emissions between the first and the last sampling event, the GHG emissions from 1st June to the first sampling and from the last sampling to 30th September were taken into consideration.

some minor corrections:

1. L77: highlevel -> high-level N deposition

Reply: We revised this part and deleted "highlevel".

2. L78: "aerobic conditions"? are the authors trying to mention redox conditions? similar expressions occur in several parts of the remaining text, consider rephrasing (e.g., L295, L383).

Reply: Yes, we meant altered redox conditions, and we revised the aerobic conditions as redox conditions in the manuscript.

3. descriptions of the mesocosm design and treatment manipulation are not very clear, the authors are recommended to include a supplementary figure for a clear illustration. in particular, the definition of $WT_0$ can be confusing (i.e., soil-water interface; L101), is it simply "the soil surface"?

Reply: We are sorry this phrase led to some confusion. In our study, the $WT_0$ means that the water level is just at the soil surface. We reformulated this explanation in the manuscript, and we added a supplementary figure to better illustrate the design as follows.

[Figure]

4. L158: despite -> regardless of. Also, the description of how the GAM is applied could be oversimplified. Ask this question may help the authors to improve the description: does the current description sufficient for peer researchers to reproduce the analysis?

Reply: We now revised the details about the application of GAM in the R listed below in L155-157. We hope this description could be clear enough to reproduce to the analysis.

Via the R package "mgcv" (Wood, 2017), we used method "gam" to perform the GAM analysis and method "predict.gam" to see the response value of GHG emissions along the N deposition gradient from 0 to 160 kg N ha$^{-1}$ yr$^{-1}$.

5. L166-168. difficult to follow. how can the heterogeneity be reduced?

Reply: We now rephrased the description about this part listed below in L159-162.

The cumulative GHG emissions in the growing seasons of each year were calculated by linear interpolation between sampling events using the trapezoidal rule (Goldberg et al., 2010). In addition to the cumulative GHG emissions between the first and the last sampling event, the GHG emissions from 1st June to the first sampling and from the last sampling to 30th September were taken into consideration.

6. L173-174. reference missed.

Reply: Thanks, but we deleted all the parts about GWP including L173-174.

7. L175. "by applying the statistic R software" -> "using R"

Reply: we now revised it in L163 as follows:

Statistical analysis was carried out using R (version 3.4.3) (R Core Team, 2017).

8. L180. SWC has been abbreviated in L146.

Reply: Thanks, we revised "Soil water content" as "SWC" in L146:

9. L205-206: "the highest value occurring"-> "with the highest value observed"

Reply: we now revised it in L192 listed below.

The response of the cumulative $CH_4$ emissions to N deposition was non-linear under $WT_0$ and $WT_{10}$ conditions (Figure 3), with the highest value observed in the $N_{20}$ treatment.

10. L245: "combination" -> "interaction"

Reply: We deleted the GWP part including this.

11. L267: add "$CH_4$" before "emissions"

Reply: we revised the whole Discussion part and deleted this part.

12. L273-274: needs rephrasing. Note that the study did not measure oxygen content, and therefore the expression like "oxygen content declined" is not appropriate. Consider: "With higher WT levels, SWC increased and likely formed more anaerobic conditions conducive to $CH_4$ production, leading to elevated $CH_4$ emissions (references)."

Reply: Thanks, and we revised the sentence listed below in L237-239.

With higher WT levels, SWC increased and likely formed more anaerobic conditions conducive to $CH_4$ production, leading to elevated $CH_4$ emissions (Evans et al., 2021; Hoyos-Santillan et al., 2019; Zhang et al., 2020).

13. L276: add comma after "considerably"

Reply: We revised this whole part and deleted the "considerably".

14. L293-295: difficult to follow.

Reply: We rephrased the part to make it clear and readable listed below in L261-264.

We speculate that the N deposition supplied N substrate and WT levels were associated with N utilization by microorganisms. Precisely, the higher WT levels promoted diffusion of the added N in the water-filled soil pore, and N thus becoming readily accessible in the microbial process to support $CH_4$ production (Wang et al., 2017).

15. L305: reference format: Gao et al. (2014).

Reply: We revised the whole paragraph in Discussion and deleted it.

16. L343: from $CH_4$ to $N_2O$ -> from $N_2O$ to $CH_4$

Reply: We deleted the whole part of GWP including "from $CH_4$ to $N_2O$".

17. L369: increased -> decreased

Reply: We revised the whole part of Section 4.3 "Implications and limitations", including "increased".

Comments 2:

Wetland is an important source of $CH_4$ and $N_2O$. Global change especially changes in precipitation and N deposition could have greatly effect on them. However, how do they affect fluxes of $CH_4$ and $N_2O$ is still unclear in wetland on the Qinghai-Tibetan Plateau. This manuscript focused on the effects of nitrogen deposition on $CH_4$ and $N_2O$ emissions under three water table levels in the Zoige alpine peatland. Thus, it is an important and interesting topic. However, there are still minor flaws that should be revised prior possible publication by this journal.

1. The present results are relying on the five levels of nitrogen deposition, but some levels (such as 160 kg N ha$^{-1}$ yr$^{-1}$) are extremely higher compared to the local nitrogen deposition (1.08-17.81 kg N ha$^{-1}$ yr$^{-1}$), could authors explain why to design the treatments?

Reply: Thanks for the comments. The reason for designing the extremely high-level N deposition in our study is because we want to consider the non-linear relationship between GHG emissions and N deposition levels, as well as to include the possible excessive N input from fertilizers or livestock excreta. We believed this high-level N deposition would not hamper us to draw a conclusion, but supporting the results even further. In fact, our main results about the relationships between the GHG emissions and N deposition are based on the N deposition levels of 0-40 kg N ha$^{-1}$ yr$^{-1}$, and the higher levels were just used to examine and confirm the relationship. We justified the design in the Section 2.3 in L105-107 listed below:

The three lowest levels ($N_0$, $N_{20}$ and $N_{40}$) cover the gradient of current and near-future deposition levels while the two highest levels ($N_{80}$ and $N_{160}$) represent levels of N-enrichment resulting from extreme deposition levels possibly combined with N input from fertilization or livestock excreta.

We also discussed the uncertainties resulting from the design related to the present results in Section 4.3 in L317-320 listed below:

It should be noted that some levels of N deposition (80 or 160 kg N ha$^{-1}$ yr$^{-1}$) in our study were much higher than the local N deposition (1.08-17.81 kg N ha$^{-1}$ yr$^{-1}$). This should not affect our general conclusion, because the non-linear and linear effects of N deposition on $CH_4$ and $N_2O$ emissions (respectively) were primarily dependent on the low levels of N deposition (0-40 kg N ha$^{-1}$ yr$^{-1}$), and the higher levels did not alter the relationship pattern.

2. Authors conducted a two-year mesocosm experiment, how about the variability of soil properties and GHG emissions within the two years. Suggest you to compare the differences of SOC, TN or GHG emissions between 2018 and 2019.

Reply: Thanks for the suggestion. We checked for the yearly differences of SOC, TN, $CH_4$ and $N_2O$ emissions between 2018 and 2019, and the figure is as follows. Unfortunately, we haven't found any clear patterns about them, and so we did not put this figure in the manuscript.

[Figure]

3. It is better to revise the second hypothesis to "The effects of N deposition on $CH_4$ and $N_2O$ emissions would be associated with WT levels" in lines 77-79.

Reply: Thanks for the comments. The second hypothesis focused on the interactive effects of N deposition and WT level on $CH_4$ and $N_2O$ emissions, but we did lack of enough data to fully support. Therefore, we deleted the two hypothesis and leave the

following two questions which could be supported by the present results, in L71-73, listed below.

In this study, we aim to adress the following two questions: i) with the N deposition consistently increasing, do the positive effects of N deposition on $CH_4$ and $N_2O$ emissions persist? ii) if there is interaction between N depostion and WT level, how do they combine to influence $CH_4$ and $N_2O$ emissions in the alpine peatland?

4. Discussion should be improved, some parts are just a repeat from the Introduction.

Reply: We have carefully and thoroughly revised the whole part of discussion.

5. English in the manuscript should be improved.

Reply: We have revised the language with help of a native speaker

Specific mistakes:

(1) delete "1% in IPCC" in the Abstract.

Reply: we deleted it.

(2) the sentence of "the large carbon pool is nitrogen deficient and is recognized …." in lines 32-33 is hard to understand and need to be rewritten.

Reply: We rephrased the sentence listed below in L30-31.

Traditionally, this nitrogen-limited ecosystem is recognized as major $CH_4$ sources and weak $N_2O$ sources (Frolking et al., 2011).

Frolking S., Talbot J., Jones M. C., Treat C. C., Kauffman J. B., Tuittila E.-S. , Roulet N.: Peatlands in the Earth's 21st century climate system, Environ. Rev., 19, 371-396. https://doi.org/10.1139/a11-014, 2011.

(2) Delete "(mean ± SE) (n=3)" in the title of table 1.

Reply: Thanks for the comments, we deleted them in the title, and put them in the table foot in L178 as follows.

Each value represents mean ± SE (n=3). SWC, soil water content; SOC, soil organic carbon; TN, total nitrogen.

(4) line 90: July should be revised to June.

Reply: We revised it in L108 as follows:

The annual added N doses were further divided into four portions and applied at the beginning of every month from June to September in 2018 and 2019.

(5) line 213: the name of Figure 1 should be changed, it is hard to see the response of GHG flux to nitrogen deposition.

Reply: we revised the name of figure 1 in L199-200 listed below.

Figure 1. Temporal variation of growing-season $CH_4$ and $N_2O$ fluxes under five levels of nitrogen deposition (0, 20, 40, 80 and 160 kg N $ha^{-1}$ $yr^{-1}$) and three water table levels in 2018 and 2019. Error bars represent the SE (n=3).

(6) Line 217, "During the rowing seasons", rowing should be revised to growing.

Reply: We revised it in L205.

Figure 2. Effects of nitrogen deposition levels on cumulative $CH_4$ and $N_2O$ emissions at three water table levels during the growing seasons in 2018 and 2019. Error bars represent the SE (n=3).

(7) Line 274: "the exposure of $CH_4$ production process to anaerobic conditions increased" might to be changed to "$CH_4$ production under anaerobic conditions was increased".

Reply: Thanks for the comments, but we rephrase it in another way in L237-239.

With higher WT levels, SWC increased and likely formed more anaerobic conditions conducive to $CH_4$ production, leading to elevated $CH_4$ emissions (Evans et al., 2021; Hoyos-Santillan et al., 2019; Zhang et al., 2020).

(8) Figure S1, the precipitations from the peatland in June, August and September of 2019 were extremely high, reaching more than 2500 mm in one month. You should scrutinize the raw data.

Reply: We recheck the original data of the precipitations from the Zoige peatland in 2019, and we found a mistake in calculating the monthly precipitation generated from the daily precipitations. We now revised it and attached the new figure S1.

[Figure]

(9) line304: "show" should be revised to "showed".

Reply: We revised this whole part including L304.

(10) line 305-306: "…the study of (Gao et al. 2014)" should be revised to "…the study of Gao et al. (2014)".

Reply: We revised this whole part and deleted the original content in L305-306.

(11) line 306-307: revise the whole sentence to "which indicated that $N_2O$ emissions was significantly increased by N addition (5.0 g N $m^{-2}$ $yr^{-1}$) and slightly decreased in the higher WT level in the alpine peatlands of the Qinghai-Tibetan Plateau."

Reply: We revised this whole paragraph and delete the original sentences in L306-307.

(12) line 313: "soil peat" should be revised to "soil".

Reply: We revised it in L277-278 as follows:

The N deposition increased soil TN (F = 4.49, $P$ = 0.002) in our study and is likely to supply more N substrate ($NH_4^+$ and $NO_3^-$) in soil (Zhu et al., 2020).

(13) line 327: (Gong et al. 2019) should be revised to Gong et al. (2019).

Reply: We thoroughly revised the Discussion and delete this part.